# Experimental evidence for recovery of mercury-contaminated fish populations

Paul J. Blanchfield[1,2,3 ✉], John W. M. Rudd[1,17], Lee E. Hrenchuk[1,3], Marc Amyot[4], Christopher L. Babiarz[5], Ken G. Beaty[1], R. A. Drew Bodaly[1], Brian A. Branfireun[6], Cynthia C. Gilmour[7], Jennifer A. Graydon[8], Britt D. Hall[9], Reed C. Harris[10], Andrew Heyes[11], Holger Hintelmann[12], James P. Hurley[13], Carol A. Kelly[1,17], David P. Krabbenhoft[14], Steve E. Lindberg[15], Robert P. Mason[16], Michael J. Paterson[1,3], Cheryl L. Podemski[1], Ken A. Sandilands[1,3], George R. Southworth[15], Vincent L. St Louis[8], Lori S. Tate[1,19] & Michael T. Tate[14]

Anthropogenic releases of mercury (Hg)[1–3] are a human health issue[4] because the potent toxicant methylmercury (MeHg), formed primarily by microbial methylation of inorganic Hg in aquatic ecosystems, bioaccumulates to high concentrations in fish consumed by humans[5,6]. Predicting the efficacy of Hg pollution controls on fish MeHg concentrations is complex because many factors influence the production and bioaccumulation of MeHg[7–9]. Here we conducted a 15-year whole-ecosystem, single-factor experiment to determine the magnitude and timing of reductions in fish MeHg concentrations following reductions in Hg additions to a boreal lake and its watershed. During the seven-year addition phase, we applied enriched Hg isotopes to increase local Hg wet deposition rates fivefold. The Hg isotopes became increasingly incorporated into the food web as MeHg, predominantly from additions to the lake because most of those in the watershed remained there. Thereafter, isotopic additions were stopped, resulting in an approximately 100% reduction in Hg loading to the lake. The concentration of labelled MeHg quickly decreased by up to 91% in lower trophic level organisms, initiating rapid decreases of 38–76% of MeHg concentration in large-bodied fish populations in eight years. Although Hg loading from watersheds may not decline in step with lowering deposition rates, this experiment clearly demonstrates that any reduction in Hg loadings to lakes, whether from direct deposition or runoff, will have immediate benefits to fish consumers.

The Minamata Convention on Mercury is an international treaty that aims to protect human health and the environment from adverse effects of MeHg by controlling Hg emissions, which should then decrease deposition and loading of anthropogenic Hg to aquatic environments[10]. Yet there is little direct evidence for how quickly fish MeHg concentrations will decline following reductions in current rates of Hg loading owing, in part, to a range of ecological factors that can influence both the microbial production and the bioaccumulation of MeHg in aquatic food webs[9,11]. Further complicating this relationship are human activities such as commercial fishing, introduction of exotic species and enhanced nutrient additions that trigger large-scale trophic disruptions[12], which can in turn substantially alter fish tissue MeHg concentrations[13–15], because fish acquire most of their MeHg through their diet[16].

Changing climatic conditions can also influence MeHg production[8], as well as restructure food webs, alter the dominant pathways of energy flow and cause size-dependent changes in fish growth rates that shift population size structure[14,17,18]. In addition, until now there has been no way to evaluate the relative contribution of newly deposited Hg to contemporary MeHg production. Consequently, it is exceedingly difficult to unambiguously assess the recovery of contaminated fish populations due specifically to Hg control measures[9].

Over a 15-year period (2001–2015) we conducted a whole-ecosystem Hg loading and recovery experiment (Mercury Experiment To Assess Atmospheric Loading In Canada and the United States (METAALICUS))[19] in a pristine boreal watershed[19]. METAALICUS addresses the relationship between changes in inorganic Hg loadings to a lake and MeHg

[1]Fisheries and Oceans Canada, Freshwater Institute, Winnipeg, Manitoba, Canada. [2]Department of Biology, Queen's University, Kingston, Ontario, Canada. [3]IISD Experimental Lakes Area, Winnipeg, Manitoba, Canada. [4]Département de Sciences Biologiques, Université de Montréal, Montreal, Quebec, Canada. [5]Environmental Chemistry and Technology Program, University of Wisconsin-Madison, Madison, WI, USA. [6]Department of Biology, Biological and Geological Sciences Building, University of Western Ontario, London, Ontario, Canada. [7]Smithsonian Environmental Research Center, Edgewater, MD, USA. [8]Department of Biological Sciences, University of Alberta, Edmonton, Alberta, Canada. [9]Department of Biology, University of Regina, Regina, Saskatchewan, Canada. [10]Reed Harris Environmental, Oakville, Ontario, Canada. [11]University of Maryland Center for Environmental Science, Chesapeake Biological Laboratory, Solomons, MD, USA. [12]Water Quality Center, Trent University, Peterborough, Ontario, Canada. [13]University of Wisconsin-Madison, Department of Civil and Environmental Engineering, Environmental Chemistry and Technology Program, Madison, WI, USA. [14]US Geological Survey, Middleton, WI, USA. [15]Oak Ridge National Laboratory, Oak Ridge, TN, USA. [16]Department of Marine Sciences, University of Connecticut, Groton, CT, USA. [17]Present address: R&K Research, Salt Spring Island, British Columbia, Canada. [18]Present address: Wisconsin Department of Natural Resources, Madison, WI, USA. ✉e-mail: Paul.Blanchfield@dfo-mpo.gc.ca

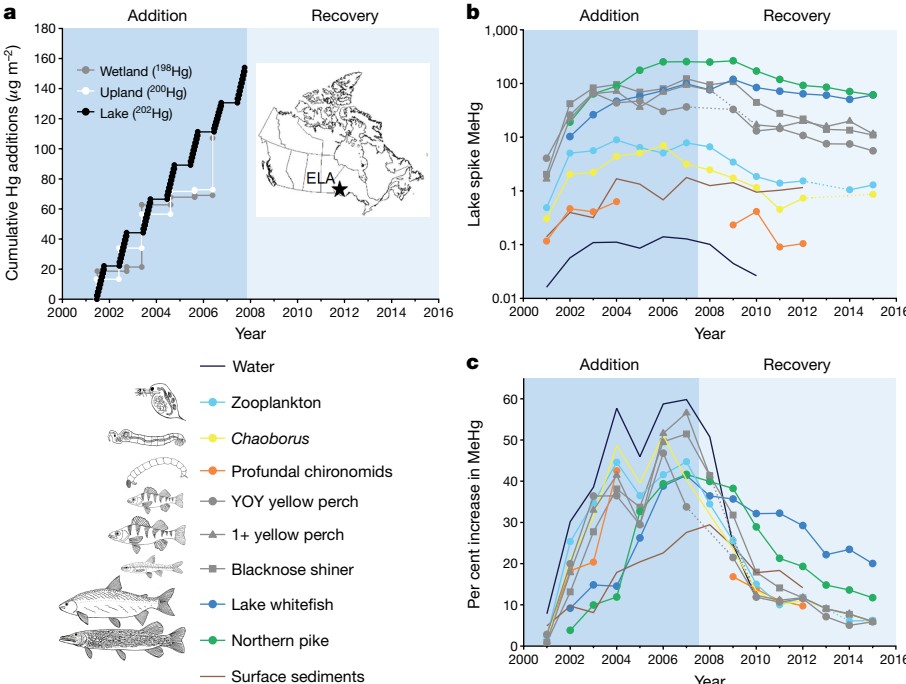

**Fig. 1 | Temporal dynamics of mercury addition and recovery in the Lake 658 ecosystem. a**, Location (inset) of the Experimental Lakes Area (ELA), Canada, where Hg enriched with different isotopes was applied to the wetland, upland and lake surface of Lake 658 to simulate enhanced wet deposition of Hg (dark blue shaded area). **b**, Inorganic Hg added to the lake was methylated and measured as MeHg concentration in water (in ng l$^{-1}$; $n = 516$), sediments (in ng g$^{-1}$ dry weight; $n = 1,627$) and invertebrates (in ng g$^{-1}$ wet weight; $n = 211$), and as total Hg in fishes (in ng g$^{-1}$ wet weight; $n = 1,052$). Mean annual concentrations

for the open-water season are shown for all lake components except for fish populations, which were collected each autumn. Concentration data for large-bodied fish are derived from body-length standardization (pike, 475 mm; whitefish, 535 mm). **c**, Hg loading to the lake increased MeHg concentrations (per cent increase = [lake spike MeHg]/[ambient MeHg] × 100) during the addition phase (2001–2007), then decreased during the recovery phase (2008–2015), when experimental Hg additions to the ecosystem ceased (light blue shaded area in **a**). Dotted lines indicate missing data.

concentrations in fish using highly enriched inorganic Hg isotopes (termed 'spikes') that enabled us to specifically follow a change in loading against a background of previously deposited Hg and present-day, relatively constant, Hg inputs from direct deposition to the lake surface and from the watershed. In our experiment, these are defined as 'ambient Hg'. By adding a different spike Hg to the lake ($^{202}$Hg), wetland ($^{198}$Hg) and upland ($^{200}$Hg) compartments of the Lake 658 watershed (52 ha in total) during a 6- to 7-year addition phase (Fig. 1a) we could follow the uptake of MeHg in fish derived solely from newly deposited Hg[19]. We then ceased all experimental additions to determine the magnitude and timing of reductions in fish MeHg concentrations to reductions in Hg loading to the lake, which we tested by tracking the decline in spike MeHg in fish, their prey and other compartments of the lake ecosystem over an eight-year recovery phase. The diverse fish community of the METAALICUS lake enabled assessment of contaminant bioaccumulation and recovery across different trophic guilds and different exposure pathways (that is, sediment versus water) for three species important to freshwater fisheries across the boreal ecoregion[20] (planktivore: yellow perch (*Perca flavescens*); benthivore: lake whitefish (*Coregonus clupeaformis*); and piscivore: northern pike (*Esox lucius*)).

The METAALICUS watershed is in an undisturbed remote region of Canada, such that our experimental addition rate increased wet Hg deposition approximately fivefold (from approximately 3.6 to 19 µg m$^{-2}$ yr$^{-1}$), to levels similar to more polluted regions of the world[21]. Most of the Hg added to the wetland and upland areas of the watershed either remained bound to vegetation and soils or evaded back to the atmosphere[22,23]. The wetland spike was below the detection level in all fish species. The Hg applied to the upland catchment accounted for only a small fraction (less than 1%) of all Hg in runoff to the lake[19] and consequently contributed little (less than 2%) to the changes in MeHg concentrations of fish populations throughout the study (Extended

Data Fig. 1). Hence, after six years of increased additions to the watershed (Fig. 1a), the observed increases in fish MeHg were due almost entirely to Hg added directly to the lake surface. This did not appear to be caused by preferential methylation of lake spike Hg. The evidence for this is that the seasonal production of lake spike MeHg, all of which had to be formed within the lake because it was added as inorganic Hg, varied synchronously with ambient MeHg in the lake and in biota[19]. This finding also indicates that in this headwater lake ambient MeHg is mainly derived from in-lake methylation of inorganic ambient Hg, most of which came from the upland catchment.

Delivery of lake spike Hg to the sediments and anoxic bottom waters, which are the dominant sites of methylation in the study lake[19,24], resulted in formation of spike MeHg in these compartments, from where spike MeHg also migrated to surface waters. Lake spike MeHg rapidly accumulated in all lake biota (Fig. 1b), with concentrations in fish muscle increasing with continued loading for all species (Extended Data Table 1; linear regression, $P < 0.05$), apart from young-of-year (YOY) yellow perch, which showed high inter-annual variability in spike MeHg concentrations after an initial increase (Fig. 1b). By contrast, ambient MeHg concentrations in all fish species did not show any consistent trends during the addition phase (Fig. 2a–c), nor did they in a nearby reference lake (Extended Data Table 1; $P > 0.05$). Steady ambient MeHg concentrations in fish through time are indicative of relatively stable watershed inputs of Hg, which is the main source of ambient inorganic Hg for methylation in both the experimental and reference lakes[19,25].

A critical question for the addition period was how much higher were MeHg concentrations than they would have been in the absence of the experimentally increased loading of Hg. The addition of lake spike Hg was roughly equivalent to all ambient Hg inputs (runoff plus direct deposition) to the lake, resulting in a doubling, or about 100% increase, in Hg loading to the lake[19]. In response to seven years of experimental

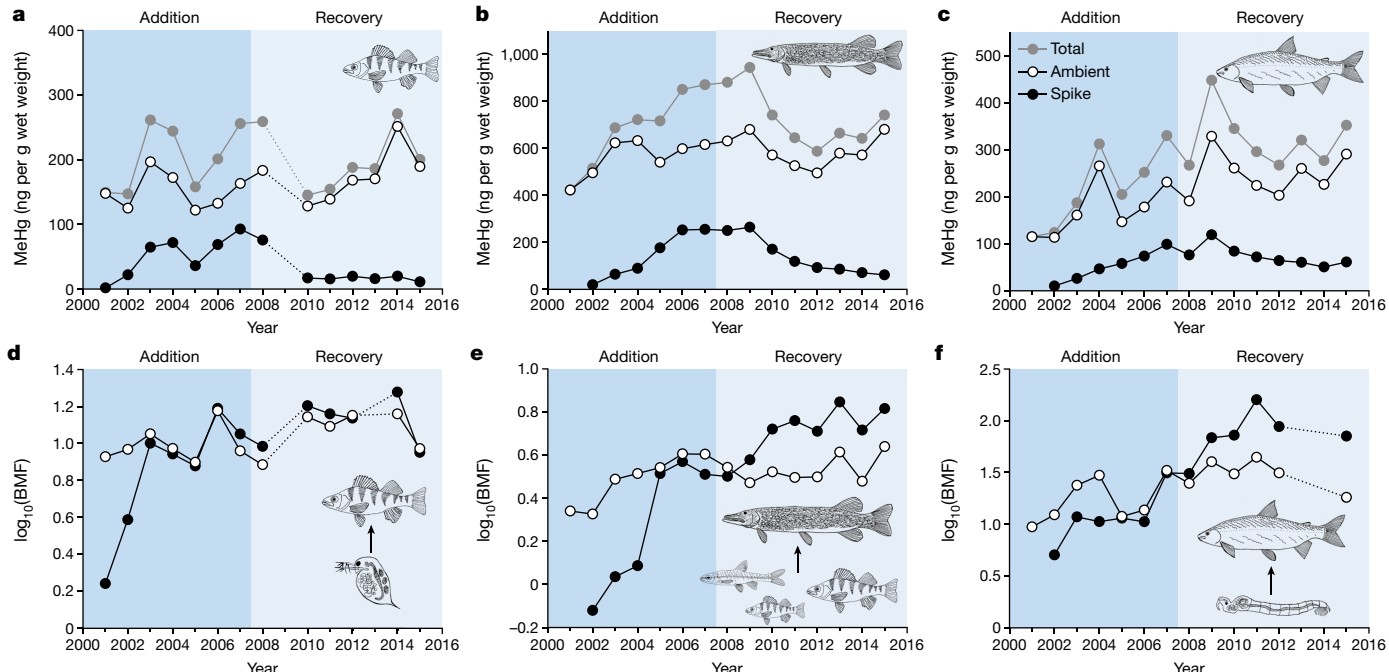

**Fig. 2 | Accumulation and trophic transfer of lake spike and ambient mercury. a–c**, Annual fish muscle MeHg concentrations (total MeHg = lake spike MeHg + ambient MeHg; grey circles) increased above background concentrations (ambient MeHg; white circles) during the addition phase (dark blue shaded area) from uptake of isotope enriched Hg added to Lake 658 (lake spike; black circles) for planktivorous (age 1+ yellow perch; $n = 140$) (**a**), piscivorous (northern pike; $n = 442$) (**b**) and benthivorous (lake whitefish; $n = 189$) (**c**) populations, then declined during the recovery phase (light blue shaded area). **d–f**, Biomagnification factors (BMF = [MeHg$_{predator}$]/[MeHg$_{prey}$]) of lake spike MeHg and ambient MeHg from dominant prey items for each of these fish species were as follows: zooplankton ($n = 127$) to yellow perch (**d**); forage fish ($n = 421$) to northern pike (**e**); and *Chaoborus* ($n = 62$) to lake whitefish (**f**). Fish data are means from autumn sampling (sample sizes in Extended Data Tables 2, 3). Concentration data for pike and whitefish are derived from body-length standardization; dotted lines indicate missing data.

additions to the lake, per cent increases in lake spike MeHg concentrations were highest in water (60%) and least in the upper 2 cm of sediments (30%) where large stores of ambient Hg existed (Fig. 1c). The response of food web organisms was intermediate to that of water and sediments, such that spike Hg additions to the lake raised MeHg concentrations by 45–57% in invertebrates and forage fishes and by more than 40% for large-bodied fish species (Fig. 1c).

Temporal patterns of biomagnification for spike MeHg relative to ambient MeHg inform how quickly the different fish species came into equilibrium with their respective prey. For small-bodied yellow perch (1 year of age) feeding on zooplankton, it took three years before the biomagnification of lake spike resembled that of ambient MeHg (Fig. 2d), and a further two years for both the apex predator, northern pike, feeding on forage fishes (Fig. 2e), and lake whitefish feeding on *Chaoborus* (Fig. 2f). Relative to planktivorous yellow perch, final addition phase concentrations of spike MeHg were slightly higher for benthivorous lake whitefish (1.2×) and further increased for piscivorous northern pike (3.9×), similar to ambient MeHg (whitefish (1.5×) and pike (3.9×); relative to perch; Fig. 2a–c) and consistent with expectations of contaminant biomagnification among trophic guilds[5,20]. These findings imply that the key in-lake processes leading to the formation and trophic transfer of MeHg to the different fish populations became comparable for spike Hg and ambient Hg during the addition phase.

To then directly test the hypothesis that MeHg concentrations in fishes would decline following reductions in Hg loading to the lake, we ceased all experimental additions of enriched Hg isotopes (Fig. 1a). This resulted in a 100% reduction in loading of lake spike. Average concentrations of lake spike MeHg in fish populations rapidly declined (within less than ten years) in concert with the decline in the availability of spike MeHg through dietary and waterborne pathways (Fig. 1b). Within the first 3 years, the relative amount of lake spike MeHg declined by 81%

in water, 35% in sediments, 66% in zooplankton and 67% in *Chaoborus* (Fig. 1c), leading to marked reductions (85–91%) in the concentration of spike MeHg in forage fish species by the end of the recovery phase (Fig. 1b). Eight years after addition, lake spike Hg contributed just a small fraction (approximately 6%) to MeHg concentrations in forage fishes and invertebrate prey (Fig. 1c). The more rapid decline in per cent spike MeHg in water compared with sediments, even in this relatively long water residence time lake (about 6 years), emphasizes that the magnitude and timing of responses by fish to Hg loading reductions could be influenced by their relative reliance upon pelagic versus benthic dietary pathways[26].

The notably fast response of the lower food web to the cessation of lake spike Hg loadings initiated rapid recovery of large-bodied fish species. Within 8 years, lake spike MeHg concentrations declined by 76% in the northern pike population and by 38% in the lake whitefish population (Fig. 1b). During the recovery phase, spike MeHg concentrations for these large-bodied species initially increased for both populations before showing steady declines (Fig. 2b, c). The rate of decline in spike MeHg for northern pike, however, was roughly twice that of lake whitefish (Fig. 1c).

Differences in the lifespan of the fish populations had a key role in the rates of recovery following the reduction of Hg loadings to the lake. Lake whitefish were much older (median age = 17 years versus 3 years for pike) and larger (Extended Data Tables 2, 3) than northern pike, and more individuals in that population would have lived through some or all of the addition and recovery phases of the experiment. Lake whitefish had the coldest thermal preferences and greatest association with benthic habitats of any fish population, which probably also contributed to their delayed recovery.

Boreal fishes are known to eliminate MeHg very slowly once accumulated[27]. To further explain the recovery of the apex predator population,

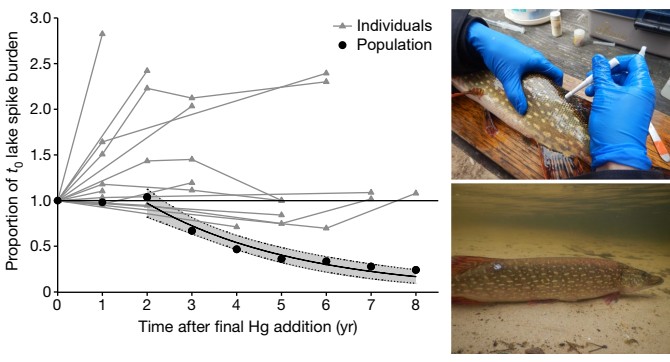

**Fig. 3 | Recovery of the apex predator from mercury loading.** Comparison of changes in body burdens of lake spike MeHg during the recovery phase for the northern pike population (annual mean, black circles) to that of individual northern pike (grey lines and triangles). Individual northern pike were sampled at the end of the addition phase (in 2007; $n = 16$) and subsequently recaptured during the recovery phase (each line represents an individual fish). Population data are based on all fish captured each autumn ($n = 280$). All northern pike were sampled using a non-lethal biopsy (represented in images) in the autumn of each year and returned to the lake. Fish body burdens of lake spike MeHg (body burden = lake spike MeHg (ng g$^{-1}$) × fish mass (g)) were normalized to concentrations in the autumn of 2007 ($t_0$; the final time isotope-enriched Hg was added to Lake 658 and the beginning of the recovery period). Exponential decay regression starting in the second year of recovery estimated a 50% reduction in lake spike MeHg burden in the population in 4.2 years (data are mean (black circle) ± 95% confidence interval (shaded band); line fit: $y = 1.7439 \times e^{-0.2928x}$, $R^2 = 0.95$, $F_{1,6} = 95.5$, $P = 0.0002$).

we tracked changes in the body burdens of spike MeHg in individual northern pike over time while also monitoring the population as a whole. As expected, individual responses were variable, but lake spike MeHg burdens in northern pike mostly increased during the early recovery phase with overall little to no loss of the spike MeHg 6–8 years after cessation of spike additions (Fig. 3). These findings parallel those observed for lake spike MeHg in individual northern pike moved from the study lake to a nearby reference lake[28] and underscore how the prolonged retention of MeHg in fish muscle tissue can delay recovery of some fisheries[7,29]. Thus, it was the annual recruitment of new fish with low MeHg concentrations into the population, along with the loss of older fish (as evidenced by a stable population size structure; Extended Data Fig. 2), that enabled the swift recovery of the population from Hg contamination as a whole. Consequently, average burdens of spike MeHg in the northern pike population were reduced by 50% in less than five years, in spite of the efficient retention of spike MeHg by some older fish (Fig. 3, Extended Data Fig. 2).

Differentiating the relative importance of present day inputs versus previously deposited Hg to overall fish MeHg concentrations is a key uncertainty when predicting the efficacy of Hg pollution reduction[29,30]. Here we demonstrate that within a few years of abatement, the experimental Hg added previously to the lake was no longer an important source of MeHg to the lower food web or to forage fishes (Fig. 1c). Long-lived fish species of subsistence, commercial and recreational importance lagged behind their prey, but the contribution of recently deposited Hg to fish MeHg steadily diminished for these populations as well. There was a similar, rapid response for the upland spike when loading ceased, even though only a small amount appeared in the fish during the experiment (Extended Data Fig. 1). The small contribution of the terrestrial spike to fish MeHg supports our former conclusion[19] that lakes with large watersheds will respond more slowly to changes in atmospheric deposition.

The most important outcome of this whole-ecosystem experiment is the demonstration that a decrease in a single factor (Hg loading to the lake) has a clear and timely effect on average MeHg concentrations in fish populations, even for long-lived species that eliminate MeHg slowly. The spike MeHg data show that fish populations will respond quickly to any change in loading rates—whether from direct deposition to the lake (Fig. 1) or runoff (Extended Data Fig. 1). Decreases in loading to the lake from these two sources will follow different time courses in response to lower atmospheric deposition[19]. However, as these two loads decrease, the fish populations in the receiving lake will soon afterwards have lower MeHg than they would have if nothing were done, thereby reducing human exposure.

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

## Methods

### Mercury additions to the study catchment

METAALICUS was conducted on the Lake 658 catchment at the Experimental Lakes Area (ELA; now IISD-ELA), a remote area in the Precambrian Shield of northwestern Ontario, Canada (49° 43′ 95″ N, 93° 44′ 20″ W) set aside for whole-ecosystem research[31]. The Lake 658 catchment includes upland (41.2 ha), wetland (1.7 ha) and lake surface (8.4 ha) areas. Lake 658 is a double basin (13 m depth), circumneutral, headwater lake, with a fish community consisting of forage (yellow perch (*P. flavescens*) and blacknose shiner (*Notropis heterolepis*)), benthivorous (lake whitefish (*C. clupeaformis*) and white sucker (*Catostomus commersonii*)), and piscivorous (northern pike (*E. lucius*)) fishes. The lake is closed to fishing.

Hg addition methods used in METAALICUS have been described in detail elsewhere[19,32,33]. In brief, three Hg spikes, each enriched with a different stable Hg isotope, were applied separately to the lake surface, upland and wetland areas. Upland and wetland spikes were applied once per year (when possible; Fig. 1a) by fixed-wing aircraft (Cessna 188 AGtruck). Mercury spikes (as $HgNO_3$) were diluted in acidified water (pH 4) in a 500 l fiberglass tank and sprayed with a stainless-steel boom on upland (approximately 79.9% $^{200}Hg$) and wetland (approximately 90.1% $^{198}Hg$) areas. Spraying was completed during or immediately before a rain event, with wind speeds less than 15 km h$^{-1}$ to minimize drift of spike Hg outside of target areas. Aerial spraying of upland and wetland areas left a 20-m buffer to the shoreline, which was sprayed by hand with a gas-powered pump and fire hose to within about 5 m of the lake[32]. Average net application rates of isotopically labelled Hg to the upland and wetland areas were 18.5 µg m$^{-2}$ yr$^{-1}$ and 17.8 µg m$^{-2}$ yr$^{-1}$, respectively.

The average net application rate for lake spike Hg was 22.0 µg m$^{-2}$ yr$^{-1}$. For each lake addition, inorganic Hg enriched with approximately 89.7% $^{202}Hg$ was added as $HgNO_3$ from four 20-l carboys filled with acidified lake water (pH 4). Nine lake additions were conducted bi-weekly at dusk over an 18-week (wk) period during the open-water season of each year (2001–2007) by injecting at 70-cm depth into the propeller wash of trolling electric motors of two boats crisscrossing each basin of the lake[32,33]. It was previously demonstrated with $^{14}C$ additions to an ELA lake that this approach evenly distributed spike added in the evening by the next morning[34].

We did not attempt to simulate Hg in rainfall for isotopic lake additions because it is impossible to simulate natural rainfall concentrations (about 10 ng l$^{-1}$) in the 20-l carboys used for additions. Instead, our starting point for the experiment was to ensure that the spike was behaving as closely as possible to ambient surface water Hg very soon after it entered the lake. Several factors support this assertion. By the next morning each spike addition had increased epilimnetic Hg concentrations by only 1 ng l$^{-1}$ $^{202}Hg$. Average ambient concentrations were 2 ng l$^{-1}$. Thus, while the Hg concentrations in the carboys were high (2.6 mg l$^{-1}$), the receiving waters were soon at trace levels. Furthermore, we investigated if the additions altered the degree of bioavailability or photoreactivity of Hg(II) in the receiving surface water. We examined the bioavailability of spike Hg(II) as compared to ambient Hg in the lake itself using a genetically engineered bioreporter bacterium[35]. On seven occasions, epilimnetic samples were collected on the day before and within 12 h of spike additions. The spike was added to the lake as Hg(NO$_3$)$_2$, which is bioavailable to the bioreporter bacterium (detection limit = 0.1 ng Hg(II) l$^{-1}$), but we never saw bioavailable ambient or spike Hg(II) in the lake, presumably because it was quickly bound to dissolved organic carbon (DOC). This indicates that, in terms of bioavailability, the spike Hg was behaving like ambient Hg soon after additions. Photoreactivity in the surface water was examined on seven occasions, by measuring the % of total Hg(II) that was dissolved gaseous Hg for spike and ambient Hg, either 24 h or 48 h after the lake was spiked[36]. There was no significant difference (paired *t*-test, *P* > 0.05),

demonstrating that by then the lake spike was behaving in the same way as ambient Hg during gaseous Hg production.

### Lake, food web and fish sampling

Water samples were collected from May to October every four weeks at the deepest point of Lake 658. Water was pumped from six depths through acid-cleaned Teflon tubing into acid-cleaned Teflon or glass bottles. Water samples were filtered in-line using pre-ashed quartz fibre filters (Whatman GFQ, 0.7 µm). Subsequently, Hg species were measured in the filtered water samples (dissolved Hg and MeHg) and in particles collected on the quartz fibre filter (particulate Hg and MeHg).

From 2001 to 2012, Lake 658 sediments were sampled at 4 fixed sites up to 5 times per year. Sampling frequency was highest in 2001, with monthly sampling from May to September, and declined over the course of the study. Fixed sites were located at depths of 0.5, 2, 3 and 7 m. A sediment survey of up to 12 additional sites was also conducted once or twice each year. Survey sites were selected to represent the full range of water depths in both basins. Cores were collected by hand by divers, or by subsampling sediments collected using a small box corer. Cores were capped and returned to the field station for processing within a few hours. For each site, three separate cores were sectioned and composited in zipper lock bags for a 0- to 2-cm depth sampling horizon, and then frozen at −20 °C.

Bulk zooplankton and *Chaoborus* samples were collected from Lake 658 for MeHg analysis. Zooplankton were collected during the day from May to October (bi-weekly: 2001–2007; monthly: 2008–2015). A plankton net (150 µm, 0.5 m diameter) was towed vertically through the water column from 1 m above the lake bottom at the deepest point to the surface of the lake. Samples were frozen in plastic Whirl-Pak bags after removal of any *Chaoborus* using acid-washed tweezers. Dominant zooplankton taxa in Lake 658 included calanoid copepods (*Diaptomus oregonensis*) and Cladocera (*Holopedium glacialis, Daphnia pulicaria* and *Daphnia mendotae*). *Chaoborus* samples were collected monthly in the same manner at least 1 h after sunset. After collection, *Chaoborus* were picked from the sample using forceps and frozen in Whirl-Pak bags. *Chaoborus* were not separated by species for MeHg analyses, but both *C. flavicans* and *C. punctipennis* occur in the lake. Profundal chironomids were sampled at the deepest part of the lake using a standard Ekman grab sampler. Grab material was washed using water from a nearby lake and individual chironomids were picked by hand.

All work with vertebrate animals was approved by Animal Care Committees (ACC) through the Canadian Council on Animal Care (Freshwater Institute ACC for Fisheries and Oceans Canada, 2001–2013; University of Manitoba ACC for IISD-ELA, 2014–2015). Licenses to Collect Fish for Scientific Purposes were granted annually by the Ontario Ministry of Natural Resources and Forestry. Prior to any Hg additions, a small-mesh fence was installed at the outlet of Lake 658 to the downstream lake to prevent movement of fish between lakes. Sampling for determination of MeHg concentrations (measured as total mercury (THg), see below) occurred each autumn (August–October; that is, the end of the growing season in north temperate lakes) for all fish species in Lake 658, and for northern pike and yellow perch in nearby reference Lake 240 (Extended Data Tables 2, 3). Fish collections occurred randomly throughout the lakes. Forage fish (YOY and 1+ yellow perch, and blacknose shiner) were captured using small mesh gillnets (6–10 mm) set for <20 min, seine nets, and hoop nets. A small number of fish (up to *n* = 20) of each species and age class (determined by visual inspection) were euthanized immediately following capture in an overdose bath of 0.25 g l$^{-1}$ tricaine methanesulfonate (TMS; Syndel Laboratories). After transport to the field station, fish were measured for fork length (FL; in mm) and mass (to 0.1 g), then immediately frozen (at −20 °C) in individual WhirlPak bags. A year class failure of yellow perch resulted in a single YOY collected in 2008 (data not presented) and no age 1+ fish in 2009 (Extended Data Table 3).

Large-bodied fish were captured by angling and multi-mesh gill nets (2.5–11.4 cm mesh) set for 20–30 min. Upon capture, each fish was anaesthetized with 0.06 g l⁻¹ TMS, measured for FL (mm), weighed (to 1 g), tagged (Passive Integrated Transponder; Biomark), and a small biopsy of dorsal muscle (0.091 ± 0.002 g wet weight (mean ± s.e.m)) was collected using a dermal punch[37]. Only fish large enough for the biopsy procedure were sampled, such that our analyses include very few juveniles (pike: 317–850 mm FL; whitefish: 344–874 mm FL). Muscle samples were inserted into 0.6-ml polypropylene vials (Rose Scientific), immediately put on ice, and frozen within 4 h (−20 °C). This non-lethal method permitted repeated sampling of individual fish over time[28,37]. The first ray of either the pectoral or pelvic fin was collected for aging purposes upon first capture. Fish recovered from anaesthesia in a tub of fresh lake water (~15 min) before being released back into the lakes. From 2001–2015, we collected 690 biopsy muscle samples from 390 fish (238 northern pike, 114 lake whitefish and 38 white sucker) in Lake 658; 149 fish (90 northern pike, 38 lake whitefish and 21 white sucker) were biopsied more than once (2 to 6 per individual). Because of consistently low annual catches of white sucker (<10 individuals) across sampling years, we have excluded them from our analyses, but note here that their patterns of lake spike MeHg accumulation and recovery were similar to those of lake whitefish. We were unable to sex most fish because they were either immature or captured outside of their spawning season.

## Sample processing and analytical methods

Detailed methods on sample preparation and MeHg or THg analysis, as well as interlaboratory calibrations, have been reported elsewhere for the METAALICUS project[19,38,39]. In brief, MeHg was distilled from water samples and from sediment using atmospheric pressure water vapour distillation and measured after aqueous phase ethylation using sodium tetraethylborate (NaBEt₄). Volatile Hg species were purged and trapped onto Tenax and MeHg was measured after thermodesorption and GC separation using inductively coupled plasma mass spectrometry (ICP-MS) detection (Micromass Platform or Perkin-Elmer Elan DRC II, respectively)[39] and quantification by species specific isotope dilution mass spectrometry. The MeHg isotope dilution standards were synthesized and calibrated in-house. Isotope-dilution spikes were added prior to distillation, and MeHg external standards were routinely calibrated against degradation by measuring the standard against inorganic Hg before and after BrCl digestion. The QC strategy include the regular analysis of blanks, laboratory duplicates and certified reference materials (CRMs) IAEA 405 (International Atomic Energy Agency, Vienna, Austria) and NIST 1566b (National Institute of Standards and Technology, Gaithersburg, Maryland) for MeHg. No CRMs are commercially available for MeHg in water.

All biota samples were handled using clean techniques with Teflon or stainless steel tools cleaned with 95% ethanol[19,38]. Zooplankton and *Chaoborus* were freeze dried, ground with an acid-washed mortar and pestle, subsampled, and weighed to the nearest 0.00001 g. For determination of MeHg concentrations (ambient, lake spike, upland spike and wetland spike) in invertebrate samples, MeHg was solubilized by treatment with a solution of KOH in ethanol (20 % w/v), ethylated by additions of NaBEt₄, and the resulting volatile Hg species were purged and trapped on carbotrap[39]. Samples were thermally desorbed and separated by gas chromatography before quantification by ICP-MS as above[39]. Samples of CRMs (TORT2 (2001–2013), IAEA452 (2014–2015); National Research Council of Canada, Ottawa, Ontario) were subjected to the same procedures; measured MeHg concentrations in the reference materials were not statistically different from certified values (*P* > 0.05).

Prey fish were kept frozen to maintain consistent wet weights. Approximately 0.2 g of skinless dorsal muscle was removed from each fish, weighed (0.0001 g), and placed in an acid-washed glass vial with a Teflon-lined cap (National Scientific Company). Muscle biopsy samples were weighed to the nearest 0.00001 g (Sartorius BP211D,

Data Weighing Systems) before and after freeze-drying (Lyph-lock 12-l freeze dry system Model 77545, Labconco) to obtain wet and dry sample masses, and dry weight proportion[28]. Fish samples were analysed for THg, which is the sum of organic and inorganic Hg. Because we had previously determined that >90% of the Hg in muscle tissue from yellow perch in Lake 658 is MeHg[40,41], here we report fish mercury data as MeHg.

THg concentrations (ambient, lake spike, upland spike and wetland spike) in fish muscle samples were quantified by ICP-MS[39]. Samples were digested with HNO₃/H₂SO₄ (7:3 v/v) and heated at 80 °C until brown NOx gases no longer formed. The THg in sample digests was reduced by SnCl₂ to Hg⁰ which was then quantified by ICP-MS (Thermo-Finnigan Element2) using a continuous flow cold vapour generation technique[41]. To correct for procedural recoveries, all samples were spiked with ²⁰¹HgCl₂ prior to sample analysis. Samples of CRMs (DORM2 (2001–2011), DORM3 (2012–2013), DORM4 (2014–2015); National Research Council of Canada) were submitted to the same procedures; measured THg concentrations in the reference materials were not statistically different from certified values (*P* > 0.05). Detection limit for each of the spikes was 0.5% of ambient Hg.

## Calculations and statistical methods

Analyses were completed with Statistica (6.1, Statsoft) and Sigmaplot (11.0, Systat Software). We present wet weight (w.w.) MeHg concentrations for all samples, except sediments which are dry weight (d.w.) concentrations. For zooplankton, *Chaoborus*, and profundal chironomids, d.w. MeHg concentrations were multiplied by a standard proportion (0.15) to yield w.w. concentrations for each sample[42]. The resulting w.w. concentrations were averaged over each open water season to determine annual means. For fish muscle biopsies, d.w. MeHg concentrations were multiplied by individual d.w. proportions to yield w.w. MeHg concentrations for each sample. To avoid any size-related biases, we calculated standardized annual MeHg concentrations (ambient and lake spike) for northern pike and lake whitefish by determining best-fit relationships between FL and MeHg concentrations for each year (quadratic polynomial, except for a linear fit for lake whitefish in 2004), and using the resulting regression equations to estimate MeHg concentrations at a standard FL[43] (the mean FL of all fish sampled for each species: northern pike, 475 mm; lake whitefish, 530 mm). Square root transformation of raw northern pike data was required to satisfy assumptions of normality and homoscedasticity prior to standardization. The resulting data represent standardized concentrations of lake spike and ambient MeHg for each species each year.

We used the ratio of lake spike and ambient Hg in each sample as a measure of the amount by which Hg concentrations were changed with the addition of isotopically enriched Hg:

$$\text{Percent increase} = [\text{lake spike Hg}]_i / [\text{ambient Hg}]_i \times 100 \qquad (1)$$

where $[\text{lake spike Hg}]_i$ is the concentration of lake spike MeHg in sample *i*, and $[\text{ambient Hg}]_i$ is the concentration of ambient MeHg in sample *i*. For northern pike and lake whitefish, we calculated the mean annual relative increase from all individuals (not the size-standardized concentration data).

Biomagnification factors (BMF) were calculated to describe differences in Hg concentrations between predator and prey[5]:

$$\text{BMF} = \log_{10}\left([\text{MeHg}]_{\text{predator}} / [\text{MeHg}]_{\text{prey}}\right) \qquad (2)$$

where $[\text{MeHg}]_{\text{predator}}$ is the mean (forage fish) or standardized (large-bodied fish) concentration of MeHg in the predator (ng g⁻¹ w.w.) and $[\text{MeHg}]_{\text{prey}}$ is the mean concentration of MeHg in the prey (ng g⁻¹ w.w.). MeHg concentration of prey items were averaged from samples collected throughout the open-water season immediately prior to autumn sampling of fish species to represent an integrated exposure for calculation of BMF. We used a dominant prey item to

represent the diet of each fish species. For age 1+ yellow perch, northern pike, and lake whitefish, dominant prey items were zooplankton, forage fishes (YOY and 1+ yellow perch, and blacknose shiner) and *Chaoborus*, respectively.

To assess loss of lake spike MeHg by northern pike during the recovery period (2008–2015), we calculated[28] whole body burdens (in µg) of lake spike MeHg for the standardized population and for individuals that had been sampled in autumn 2007 ($t_0$ is the final time spike Hg was added to the lake) and again in at least one subsequent year during annual autumn sampling ($n$ = 16 fish, of which 1–9 individuals were recaptured annually from 2008–2015). This calculation of MeHg burden is a relative measure of whole fish Hg content because MeHg is higher in muscle tissue than in other tissue types[28,40]. For the standardized population data, we used best-fit relationships between FL (in mm) and body weight (in g; quadratic polynomial) to determine body weight at the standard FL. We multiplied this body weight by standard ambient and spike MeHg concentrations (in ng g$^{-1}$ w.w.) in muscle tissue for each year to determine body burdens over time (in ng). For individual fish, we multiplied spike MeHg concentration (in ng g$^{-1}$ w.w.) by body weight (in g) to yield individual body burdens (in ng). To account for differences among individuals and between individuals and the population, we normalized the data to examine the mean proportion of original ($t_0$) lake spike MeHg burden present in northern pike each year of the recovery period (2008–2015).

$$\text{change in burden from } t_0 = \text{burden}_{tx}/\text{burden}_{t0} \quad (3)$$

We used a best fit regression (exponential decay, beginning in the second year of recovery) to estimate the half-life (50% of original burden) of lake spike MeHg for the population.

Northern pike and lake whitefish ages were determined by cleithra and otoliths, respectively, if mortality had occurred, but most ages were quantified using fin rays collected from live fish[44] (K. H. Mills, DFO or North/South Consultants). Northern pike of the sizes selected for biopsy sampling had a median age of 3 years (range: 2–12 years; $n$ = 305); the median age of lake whitefish was 17 years (range: 3–38 years; $n$ = 86).

## Reporting summary

Further information on research design is available in the Nature Research Reporting Summary linked to this paper.

## Data availability

Datasets generated in this study are available at https://doi.org/10.5061/dryad.nzs7h44sf. Source data are provided with this paper.

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

**Acknowledgements** We thank A. Robinson for aircraft support, M. Stainton and R. Hesslein for the preparation of mercury isotopes, M. Lyng, J. T. Bell, R. Burrell, B. Dimock, M. Dobrin, K. Fetterly, C. Miller, M. Rearick, J. Reid, G. Riedel, J. Shead, J. Zhou and staff and students at the Experimental Lakes Area for assistance with field collections and analysis of samples. K. Kidd, D. Orihel and J. Smol provided reviews of this manuscript. We thank the following agencies for their financial support for this study: Electric Power Research Institute, Environment and Climate Change Canada, Fisheries and Oceans Canada, Natural Sciences and Engineering Research Council of Canada, National Science Foundation Grant DEB 0451345 (C.C.G.) and DEB 0351050 (A.H. and C.C.G.), Southern Company, University of Alberta, U.S. Department of Energy, U.S. Environmental Protection Agency, and Wisconsin Focus on Energy Program Project Grant no. 4900-02-03 (J.P.H.). Any use of trade, firm, or product names is for descriptive purposes only and does not imply endorsement by the U.S. Government.

**Author contributions** P.J.B., J.W.M.R., M.A., K.G.B., R.A.B., B.A.B., C.C.G., R.C.H., A.H., H.H., J.P.H., C.A.K., D.P.K., S.E.L., R.P.M., M.J.P., C.L.P. and V.L.S.L. contributed to the design of the whole-ecosystem experiment. The METAALICUS project was overseen by J.W.M.R. and R.C.H., who along with C.L.B., R.A.B., J.A.G., H.H., J.P.H., C.A.K., D.P.K., K.A.S., V.L.S.L. and M.T.T., applied mercury to the lake and watershed. P.J.B., L.E.H. and L.S.T. conducted the fish sampling. Water, sediment and lower trophic level data were collected and prepared by C.C.G., B.D.H., H.H., C.A.K., M.J.P., C.L.P. and K.A.S. Field samples in this study were analysed for mercury by C.C.G., H.H., D.P.K. and M.T.T. Data analyses were performed by P.J.B. and L.E.H. All authors collected and discussed project-level data that contributed to the interpretation of the data presented in this study. L.E.H. produced the figures and wrote the methods. P.J.B., J.W.M.R., C.A.K., and V.L.S.L. wrote the manuscript with input from all authors.

**Competing interests** The authors declare no competing interests.

**Additional information**
**Correspondence and requests for materials** should be addressed to Paul J. Blanchfield.

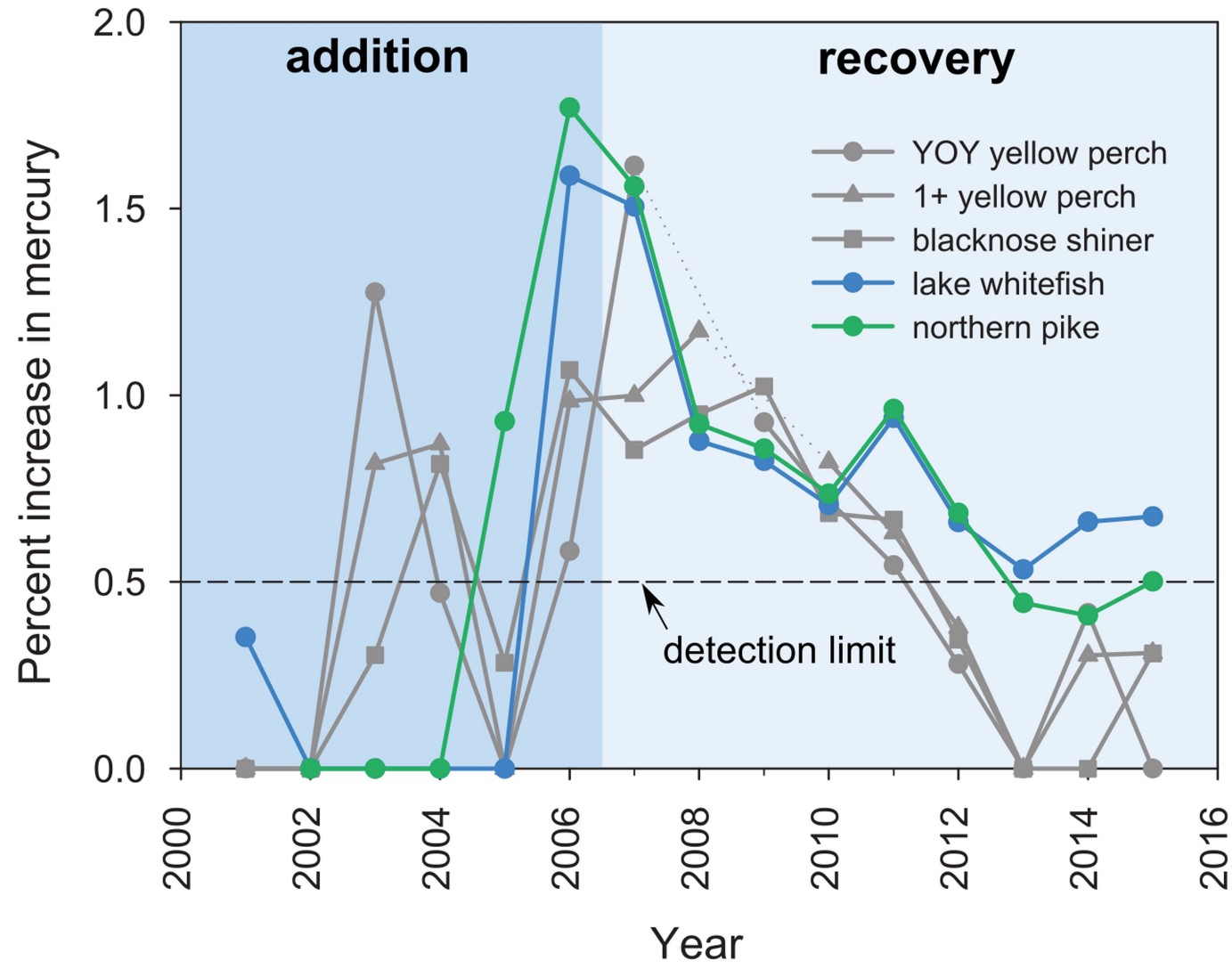

**Extended Data Fig. 1 | Temporal dynamics of upland mercury in fish.** Isotopic $^{198}$Hg added to the upland area of Lake 658 was above the detection limit (0.5% of ambient MeHg; dashed line) in all fish species, but contributed little (<2%) to overall MeHg concentrations (percent increase = [upland spike MeHg]/[ambient MeHg] × 100). Mean annual MeHg concentration data for each species or age-class is presented and based on fish collected during fall population sampling ($n$ = 1,052; sample size details in Extended Data Tables 2 and 3); dotted lines indicate missing data.

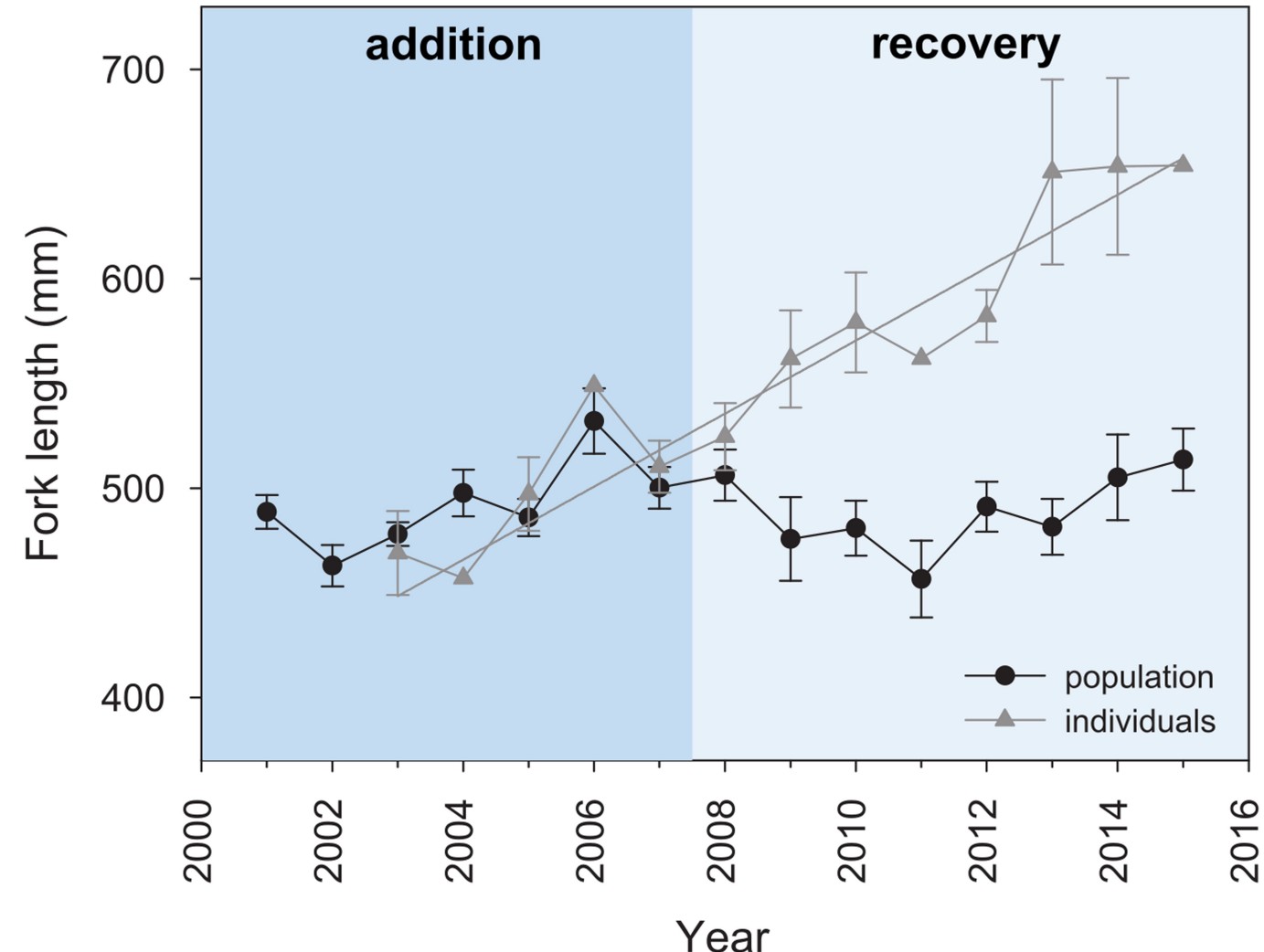

**Extended Data Fig. 2 | Comparison of individual and population body sizes of northern pike.** Mean (± s.e.m.) body size of all northern pike sampled in the fall of each year (population, black circles; $n$ = 442) for muscle MeHg concentration using a biopsy method was stable over time. Individual northern pike (grey triangles) captured in 2007 and again in at least one subsequent year ($n$ = 16 fish with 1–9 individuals recaptured each year 2008–2015) were used to determine individual losses of lake spike MeHg during recovery (see Fig. 3). These individual fish, which were also captured prior to 2007, showed an increase in body size over time (linear regression: $y = 17.43x - 34470.0$, $R^2$ = 0.55, $F_{1,51}$ = 61.1, $P < 0.0001$).

**Extended Data Table 1 | Fish mercury concentrations in the experimental and reference lakes over time**

| lake | species | mercury type | addition (2001-2007) | | | | all years (2001-2015) | | | |
|------|---------|--------------|------|------|------|------|------|------|------|------|
| | | | y0 | a | R² | P | y0 | a | R² | P |
| 658 | 0+ yellow perch | ambient | 19081.7 | -9.5 | 0.28 | 0.2266 | 5250.0 | -2.6 | 0.14 | 0.1769 |
| | | lake spike | -6148.2 | 3.1 | 0.12 | 0.4553 | - | | | |
| | 1+ yellow perch | ambient | 1094.4 | -0.5 | 0.001 | 0.9375 | -6935.5 | 3.5 | 0.21 | 0.0997 |
| | | lake spike | -24109.9 | 12.1 | 0.66 | 0.0261 | - | | | |
| | blacknose shiner | ambient | 31759.2 | -15.7 | 0.38 | 0.1393 | 15679.5 | -7.7 | 0.35 | 0.0201 |
| | | lake spike | -30228.2 | 15.1 | 0.70 | 0.0185 | - | | | |
| | northern pike | ambient | -49814.4 | 25.1 | 0.47 | 0.0872 | -9877.2 | 5.2 | 0.11 | 0.2398 |
| | | lake spike* | -104618.5 | 52.3 | 0.96 | 0.0007 | - | | | |
| | lake whitefish | ambient | -32983.6 | 16.5 | 0.39 | 0.1344 | -18755.1 | 9.4 | 0.45 | 0.0063 |
| | | lake spike* | -34096.6 | 17.0 | 0.99 | <0.0001 | - | | | |
| 240 | 0+ yellow perch | ambient | 1205.5 | -0.6 | 0.003 | 0.8939 | -464.5 | 0.3 | 0.003 | 0.8947 |
| | 1+ yellow perch | ambient | 17990.7 | -8.9 | 0.15 | 0.3880 | 2722.7 | -1.3 | 0.03 | 0.5428 |
| | northern pike | ambient | 38790.9 | -19.0 | 0.24 | 0.2691 | 33611.0 | -16.4 | 0.36 | 0.0180 |

Footnote: Linear regression statistics [$F = y0 + (a \times x)$] of the change in annual mean (yellow perch, blacknose shiner) or body-size standardized (northern pike, lake whitefish) concentrations (ng g$^{-1}$ w.w.) of ambient and lake spike MeHg (measured as THg) in fish muscle tissue during the addition phase or entire study. Isotopic Hg additions were made to Lake 658 from 2001–2007; Lake 240 is a reference lake. Fish collections occurred each autumn.
*lake spike below detection in 2001 (pike: $n = 32$; whitefish: $n = 7$ of 9); regression from 2002–2007

## Extended Data Table 2 | Annual fish metrics and mercury concentrations during the addition phase

| lake | species | parameter | 2001 | 2002 | 2003 | 2004 | 2005 | 2006 | 2007 |
|---|---|---|---|---|---|---|---|---|---|
| 658 | 0+ yellow perch | n | 12 | 15 | 8 | 10 | 10 | 10 | 10 |
| | | weight (g) | 2.2 (0.1) | 1.4 (0.1) | 3.2 (0.2) | 1.4 (0.04) | 0.7 (0.04) | 2.1 (0.2) | 2.5 (0.2) |
| | | fork length (mm) | 61 (1) | 51 (0.6) | 65 (1.5) | 50 (0.3) | 39 (0.7) | 53 (1.6) | 60 (2.1) |
| | | ambient THg (ng.g$^{-1}$ w.w.) | 146 (10) | 129 (3) | 185 (14) | 119 (9) | 161 (7) | 65 (5) | 109 (3) |
| | | lake spike THg (ng.g$^{-1}$ w.w.) | 4 (0.3) | 26 (0.7) | 67 (4.5) | 44 (3.8) | 47 (1.8) | 30 (2.1) | 36 (1.1) |
| | | percent lake spike THg | 3 (0.2) | 20 (0.2) | 36 (0.4) | 36 (0.6) | 29 (0.2) | 47 (0.4) | 34 (1.3) |
| 658 | 1+ yellow perch | n | 9 | 6 | 10 | 10 | 10 | 15 | 10 |
| | | weight (g) | 4.9 (0.3) | 3.9 (0.3) | 7.5 (0.4) | 5.1 (0.5) | 2 (0.2) | 4.5 (0.4) | 7.5 (0.4) |
| | | fork length (mm) | 78 (1.7) | 72 (2.1) | 88 (1.5) | 75 (2.1) | 55 (1.2) | 73 (2) | 88 (1.7) |
| | | ambient THg (ng.g$^{-1}$ w.w.) | 148 (17) | 125 (14) | 197 (13) | 172 (12) | 122 (10) | 132 (18) | 163 (10) |
| | | lake spike THg (ng.g$^{-1}$ w.w.) | 2 (0.3) | 22 (1.9) | 65 (3.9) | 72 (5.3) | 36 (3) | 69 (9.3) | 93 (6) |
| | | percent lake spike THg | 1 (0.2) | 18 (0.8) | 33 (0.8) | 41 (0.6) | 30 (0.6) | 52 (0.3) | 57 (0.7) |
| 658 | blacknose shiner | n | 8 | 15 | 10 | 10 | 10 | 11 | 4 |
| | | weight (g) | 1.2 (0.2) | 1.7 (0.1) | 1.8 (0.04) | 1.7 (0.1) | 1.6 (0.1) | 1.4 (0.1) | 1.6 (0.03) |
| | | fork length (mm) | 50 (2.9) | 56 (0.5) | 55 (0.6) | 55 (0.8) | 52 (1.2) | 52 (1.7) | 54 (0.5) |
| | | ambient THg (ng.g$^{-1}$ w.w.) | 245 (30) | 322 (12) | 303 (19) | 254 (15) | 202 (25) | 162 (17) | 238 (5) |
| | | lake spike THg (ng.g$^{-1}$ w.w.) | 2 (0.3) | 42 (1.8) | 83 (4) | 96 (5) | 69 (9) | 80 (8) | 123 (4) |
| | | percent lake spike THg | 1 (0.2) | 13 (0.5) | 28 (0.9) | 38 (1) | 34 (0.7) | 50 (0.4) | 51 (0.8) |
| 658 | lake whitefish | n | 9 | 17 | 6 | 7 | 9 | 13 | 8 |
| | | weight (g) | 1928 (220) | 2498 (86) | 2448 (132) | 2598 (170) | 2853 (106) | 2739 (190) | 2190 (272) |
| | | fork length (mm) | 510 (24) | 543 (7) | 542 (10) | 558 (9) | 566 (8) | 551 (18) | 527 (23) |
| | | ambient THg (ng.g$^{-1}$ w.w.)* | 115 (42-156) | 114 (44-184) | 161 (84-240) | 265 (156-595)‡ | 147 (104-226) | 178 (118-276) | 231 (201-791) |
| | | lake spike THg (ng.g$^{-1}$ w.w.)* | nd† | 10 (5-15) | 26 (14-39) | 47 (25-56)‡ | 58 (28-64) | 74 (42-104) | 99 (69-349) |
| | | percent lake spike THg | nd† | 9 (0.6) | 15 (2.1) | 15 (1) | 26 (2.8) | 39 (2.5) | 41 (3.4) |
| 658 | northern pike | n | 32 | 24 | 37 | 29 | 23 | 17 | 30 |
| | | weight (g) | 705 (30) | 650 (39) | 695 (24) | 1051 (131) | 788 (43) | 969 (94) | 837 (51) |
| | | fork length (mm) | 489 (8) | 463 (10) | 478 (6) | 532 (19) | 486 (9) | 532 (16) | 500 (10) |
| | | ambient THg (ng.g$^{-1}$ w.w.)* | 422 (130-1430) | 495 (213-647) | 623 (346-999) | 632 (325-2163) | 540 (307-828) | 598 (484-755) | 616 (257-1355) |
| | | lake spike THg (ng.g$^{-1}$ w.w.)* | nd | 19 (1-37) | 64 (14-84) | 89 (0-208) | 176 (102-236) | 252 (147-286) | 255 (117-514) |
| | | percent lake spike THg | nd | 4 (0.5) | 10 (0.6) | 12 (1.3) | 33 (1.3) | 39 (1.6) | 42 (0.9) |
| 240 | 0+ yellow perch | n | 10 | 15 | 10 | 10 | 9 | 10 | 10 |
| | | weight (g) | 2.0 (0.1) | 1.7 (0.1) | 2 (0.1) | 0.7 (0.1) | 1.0 (0.1) | 0.9 (0.1) | 1.5 (0.3) |
| | | fork length (mm) | 59 (1.2) | 56 (0.6) | 63 (1.2) | 40 (1.5) | 48 (1.5) | 47 (1.4) | 54 (1.0) |
| | | ambient THg (ng.g$^{-1}$ w.w.) | 125 (4) | 82 (4) | 48 (4) | 56 (2) | 82 (4) | 74 (5) | 98 (7) |
| 240 | 1+ yellow perch | n | 4 | 10 | 10 | 10 | 11 | 10 | 10 |
| | | weight (g) | 9.6 (1.5) | 5.0 (0.3) | 7.3 (0.6) | 8.0 (0.6) | 4.2 (0.4) | 4.1 (0.3) | 5.6 (0.3) |
| | | fork length (mm) | 104 (5) | 81 (1.4) | 93 (2.7) | 94 (2) | 75 (2) | 74 (1.5) | 82 (1.8) |
| | | ambient THg (ng.g$^{-1}$ w.w.) | 199 (26) | 150 (5) | 131 (8) | 179 (16) | 66 (3) | 89 (5) | 81 (5) |
| 240 | northern pike | n | 17 | 8 | 20 | 26 | 21 | 10 | 20 |
| | | weight (g) | 2424 (199) | 2016 (211) | 1896 (141) | 1618 (122) | 1597 (106) | 2066 (167) | 1950 (123) |
| | | fork length (mm) | 675 (23) | 686 (34) | 631 (19) | 608 (18) | 591 (16) | 656 (24) | 649 (16) |
| | | ambient THg (ng.g$^{-1}$ w.w.)* | 725 (408-1763) | 856 (423-2638) | 739 (370-1574) | 715 (242-1629) | 612 (44-2033) | 614 (234-1178) | 752 (186-1346) |

Footnote: Data presented are mean (± s.e.m.) of $n$ fish measured and their muscle tissue analysed for ambient and lake spike MeHg concentrations (ng g$^{-1}$ w.w., measured as THg), except for northern pike and lake whitefish where concentration data were standardized by fish length (range in parentheses). Percent lake spike THg = [lake spike THg]$_i$/[ambient THg]$_i$ × 100. Isotopic Hg additions were made to Lake 658 from 2001–2007; Lake 240 is a reference lake. Fish collections occurred each autumn.

nd = not detected (detection limit for lake spike is 0.5% of ambient THg).

*Standardized by fish length (pike: 475 mm; whitefish: 535 mm).

†Two fish above detection limit.

‡Standardized with linear regression instead of polynomial.

**Extended Data Table 3 | Annual fish metrics and mercury concentrations during the recovery phase**

| lake | species | parameter | 2008 | 2009 | 2010 | 2011 | 2012 | 2013 | 2014 | 2015 |
|---|---|---|---|---|---|---|---|---|---|---|
| 658 | 0+ yellow | n | 0* | 10 | 10 | 10 | 10 | 10 | 10 | 10 |
| | perch | weight (g) | | 1.7 (0.1) | 1.5 (0.1) | 1.9 (0.1) | 1.8 (0.1) | 2.0 (0.1) | 1.9 (0.2) | 1.7 (0.1) |
| | | fork length (mm) | | 51 (0.8) | 51 (1.4) | 54 (0.9) | 55 (1.2) | 56 (0.9) | 55 (1.7) | 53 (0.8) |
| | | ambient THg (ng·g$^{-1}$ w.w.) | | 153 (9) | 110 (5) | 136 (5) | 93 (7) | 105 (4) | 143 (15) | 95 (9) |
| | | lake spike THg (ng.g$^{-1}$ w.w.) | | 33 (1.9) | 13 (0.7) | 15 (0.6) | 11 (0.9) | 8 (0.3) | 7 (1.1) | 6 (0.5) |
| | | percent lake spike THg | | 21 (0.1) | 12 (0.2) | 11 (0.03) | 12 (0.2) | 7 (0.1) | 5 (0.3) | 6 (0.1) |
| 658 | 1+ yellow | n | 10 | 0* | 10 | 10 | 10 | 10 | 10 | 10 |
| | perch | weight (g) | 4.2 (0.2) | | 3.7 (0.2) | 3.9 (0.3) | 6.1 (0.4) | 5.1 (0.4) | 5.2 (0.4) | 4.2 (0.3) |
| | | fork length (mm) | 72 (1.2) | | 68 (1.6) | 70 (1.6) | 82 (2.1) | 77 (2.4) | 79 (2.3) | 74 (1.8) |
| | | ambient THg (ng.g$^{-1}$ w.w.) | 183 (6) | | 128 (16) | 139 (9) | 168 (14) | 170 (19) | 251 (23) | 195 (19) |
| | | lake spike THg (ng.g$^{-1}$ w.w.) | 76 (2.1) | | 17 (3.7) | 15 (1) | 20 (1.6) | 16 (2) | 20 (2) | 12 (1.1) |
| | | percent lake spike THg | 41 (0.4) | | 12 (1.1) | 11 (0.1) | 12 (0.1) | 9 (0.5) | 8 (0.4) | 6 (0.04) |
| 658 | blacknose | n | 10 | 10 | 10 | 10 | 3 | 5 | 10 | 10 |
| | shiner | weight (g) | 1.7 (0.1) | 1.5 (0.1) | 1.2 (0.1) | 1.9 (0.1) | 1.9 (0.01) | 1.6 (0.1) | 1.7 (0.1) | 1.8 (0.1) |
| | | fork length (mm) | 54 (1) | 52 (0.5) | 50 (1.2) | 56 (1) | 57 (0.3) | 52 (1) | 55 (0.8) | 56 (1) |
| | | ambient THg (ng.g$^{-1}$ w.w.) | 228 (12) | 340 (17) | 238 (17) | 189 (22) | 185 (28) | 154 (7) | 176 (9) | 178 (20) |
| | | lake spike THg (ng.g$^{-1}$ w.w.) | 95 (7) | 108 (6) | 44 (5) | 28 (4) | 22 (4) | 14 (0.7) | 14 (0.7) | 11 (1.6) |
| | | percent lake spike THg | 41 (1) | 32 (1) | 18 (1) | 14 (1) | 12 (0.4) | 9 (0.1) | 8 (0.2) | 6 (0.3) |
| 658 | lake | n | 8 | 17 | 11 | 13 | 20 | 10 | 19 | 22 |
| | whitefish | weight (g) | 2357 (276) | 2469 (113) | 1938 (203) | 2208 (130) | 2197 (74) | 2097 (91) | 2243 (137) | 2136 (136) |
| | | fork length (mm) | 540 (25) | 550 (8) | 495 (22) | 571 (27) | 532 (7) | 527 (9) | 535 (13) | 523 (13) |
| | | ambient THg (ng.g$^{-1}$ w.w.)† | 191 (101-341) | 328 (165-480) | 261 (180-394) | 224 (94-404) | 203 (128-346) | 260 (143-469) | 226 (144-506) | 291 (155-439) |
| | | lake spike THg (ng.g$^{-1}$ w.w.)† | 76 (24-102) | 119 (61-165) | 84 (49-134) | 72 (36-125) | 64 (38-96) | 60 (21-140) | 51 (17-106) | 61 (15-111) |
| | | percent lake spike THg | 36 (4) | 36 (1.4) | 32 (1.2) | 34 (1.1) | 29 (0.6) | 22 (1.2) | 23 (1.4) | 20 (1.2) |
| 658 | northern | n | 27 | 17 | 39 | 29 | 38 | 46 | 29 | 25 |
| | pike | weight (g) | 837 (64) | 704 (85) | 784 (62) | 667 (81) | 822 (58) | 769 (71) | 901 (149) | 886 (76) |
| | | fork length (mm) | 506 (12) | 476 (20) | 481 (13) | 457 (18) | 491 (12) | 482 (13) | 497 (20) | 514 (15) |
| | | ambient THg (ng.g$^{-1}$ w.w.)† | 631 (338-1279) | 680 (380-1575) | 571 (279-1180) | 526 (212-1723) | 495 (102-1119) | 579 (298-1903) | 571 (255-1661) | 680 (530-1513) |
| | | lake spike THg (ng.g$^{-1}$ w.w.)† | 250 (29-480) | 264 (133-642) | 170 (31-454) | 119 (34-633) | 92 (26-323) | 85 (32-494) | 71 (17-445) | 61 (40-390) |
| | | percent lake spike THg | 40 (1.3) | 38 (0.5) | 29 (1.2) | 21 (1.4) | 19 (1) | 15 (0.7) | 13 (1.1) | 12 (0.9) |
| 240 | 0+ yellow | n | 11 | 15 | 10 | 10 | 10 | 10 | 10 | 10 |
| | perch | weight (g) | 1.1 (0.1) | 0.7 (0.1) | 1.6 (0.2) | 1.4 (0.2) | 2.6 (0.1) | 2.6 (0.2) | 1.6 (0.1) | 2.2 (0.2) |
| | | fork length (mm) | 48 (1.5) | 42 (1.0) | 52 (1.7) | 51 (1.9) | 65 (0.8) | 62 (1.7) | 54 (1.7) | 59 (2.0) |
| | | ambient THg (ng.g$^{-1}$ w.w.) | 91 (5) | 91 (4) | 109 (6) | 100 (10) | 83 (3) | 52 (10) | 110 (4) | 65 (3) |
| 240 | 1+ yellow | n | 10 | 10 | 10 | 10 | 10 | 10 | 10 | 9 |
| | perch | weight (g) | 3.6 (0.1) | 3.4 (0.3) | 4.6 (0.3) | 4.2 (0.2) | 6.2 (0.2) | 7.5 (0.4) | 6.6 (0.4) | 4.8 (0.5) |
| | | fork length (mm) | 71 (0.7) | 69 (1.7) | 77 (2) | 73 (1.2) | 86 (1.3) | 89 (1.7) | 87 (1.3) | 77 (3) |
| | | ambient THg (ng.g$^{-1}$ w.w.) | 154 (10) | 118 (11) | 145 (16) | 126 (6) | 150 (11) | 127 (3) | 140 (8) | 79 (7) |
| 240 | northern | n | 20 | 7 | 18 | 14 | 19 | 15 | 16 | 23 |
| | pike | weight (g) | 1933 (90) | 1991 (202) | 1482 (113) | 1290 (128) | 1797 (151) | 1845 (129) | 2150 (159) | 1800 (172) |
| | | fork length (mm) | 650 (16) | 659 (34) | 582 (15) | 557 (18) | 618 (19) | 619 (16) | 644 (17) | 600 (19) |
| | | ambient THg (ng.g$^{-1}$ w.w.)† | 817 (318-2058) | 771 (397-1504) | 750 (327-1343) | (207-1811) | 445 (134-826) | 644 (292-1999) | 530 (342-1312) | 584 (195-1449) |

Footnote: Data presented are mean (± s.e.m.) of $n$ fish measured and their muscle tissue analysed for ambient and lake spike MeHg concentrations (ng g$^{-1}$ w.w., measured as THg), except for northern pike and lake whitefish where concentration data were standardized by fish length (range in parentheses). Percent lake spike THg = [lake spike THg]$_i$/[ambient THg]$_i$ × 100. Isotopic Hg additions were made to Lake 658 from 2001–2007; Lake 240 is a reference lake. Fish collections occurred each autumn.
*Year class failure.
†Standardized by fish length (pike: 475 mm; whitefish: 535 mm).

# Reporting Summary

## Statistics

For all statistical analyses, confirm that the following items are present in the figure legend, table legend, main text, or Methods section.

| n/a | Confirmed | |
|---|---|---|
| ☐ | ☒ | The exact sample size (*n*) for each experimental group/condition, given as a discrete number and unit of measurement |
| ☐ | ☒ | A statement on whether measurements were taken from distinct samples or whether the same sample was measured repeatedly |
| ☐ | ☒ | The statistical test(s) used AND whether they are one- or two-sided<br>*Only common tests should be described solely by name; describe more complex techniques in the Methods section.* |
| ☒ | ☐ | A description of all covariates tested |
| ☐ | ☒ | A description of any assumptions or corrections, such as tests of normality and adjustment for multiple comparisons |
| ☐ | ☒ | A full description of the statistical parameters including central tendency (e.g. means) or other basic estimates (e.g. regression coefficient) AND variation (e.g. standard deviation) or associated estimates of uncertainty (e.g. confidence intervals) |
| ☐ | ☒ | For null hypothesis testing, the test statistic (e.g. *F*, *t*, *r*) with confidence intervals, effect sizes, degrees of freedom and *P* value noted<br>*Give P values as exact values whenever suitable.* |
| ☒ | ☐ | For Bayesian analysis, information on the choice of priors and Markov chain Monte Carlo settings |
| ☒ | ☐ | For hierarchical and complex designs, identification of the appropriate level for tests and full reporting of outcomes |
| ☒ | ☐ | Estimates of effect sizes (e.g. Cohen's *d*, Pearson's *r*), indicating how they were calculated |

*Our web collection on statistics for biologists contains articles on many of the points above.*

## Software and code

Policy information about availability of computer code

| Data collection | *No software was used.* |
|---|---|
| Data analysis | *Statistica v 6.1 (Statsoft Inc.); SigmaPlot v11.0 (Systat Software Inc.); Microsoft Excel (2010)* |

For manuscripts utilizing custom algorithms or software that are central to the research but not yet described in published literature, software must be made available to editors and reviewers. We strongly encourage code deposition in a community repository (e.g. GitHub). See the Nature Portfolio guidelines for submitting code & software for further information.

## Data

Policy information about availability of data

All manuscripts must include a data availability statement. This statement should provide the following information, where applicable:

- Accession codes, unique identifiers, or web links for publicly available datasets
- A description of any restrictions on data availability
- For clinical datasets or third party data, please ensure that the statement adheres to our policy

*Datasets generated in this study are available at (doi:10.5061/dryad.nzs7h44sf) and are provided as Source data.*

# Field-specific reporting

Please select the one below that is the best fit for your research. If you are not sure, read the appropriate sections before making your selection.

☐ Life sciences          ☐ Behavioural & social sciences          ☒ Ecological, evolutionary & environmental sciences

For a reference copy of the document with all sections, see nature.com/documents/nr-reporting-summary-flat.pdf

# Life sciences study design

All studies must disclose on these points even when the disclosure is negative.

| | |
|---|---|
| Sample size | *Describe how sample size was determined, detailing any statistical methods used to predetermine sample size OR if no sample-size calculation was performed, describe how sample sizes were chosen and provide a rationale for why these sample sizes are sufficient.* |
| Data exclusions | *Describe any data exclusions. If no data were excluded from the analyses, state so OR if data were excluded, describe the exclusions and the rationale behind them, indicating whether exclusion criteria were pre-established.* |
| Replication | *Describe the measures taken to verify the reproducibility of the experimental findings. If all attempts at replication were successful, confirm this OR if there are any findings that were not replicated or cannot be reproduced, note this and describe why.* |
| Randomization | *Describe how samples/organisms/participants were allocated into experimental groups. If allocation was not random, describe how covariates were controlled OR if this is not relevant to your study, explain why.* |
| Blinding | *Describe whether the investigators were blinded to group allocation during data collection and/or analysis. If blinding was not possible, describe why OR explain why blinding was not relevant to your study.* |

# Behavioural & social sciences study design

All studies must disclose on these points even when the disclosure is negative.

| | |
|---|---|
| Study description | *Briefly describe the study type including whether data are quantitative, qualitative, or mixed-methods (e.g. qualitative cross-sectional, quantitative experimental, mixed-methods case study).* |
| Research sample | *State the research sample (e.g. Harvard university undergraduates, villagers in rural India) and provide relevant demographic information (e.g. age, sex) and indicate whether the sample is representative. Provide a rationale for the study sample chosen. For studies involving existing datasets, please describe the dataset and source.* |
| Sampling strategy | *Describe the sampling procedure (e.g. random, snowball, stratified, convenience). Describe the statistical methods that were used to predetermine sample size OR if no sample-size calculation was performed, describe how sample sizes were chosen and provide a rationale for why these sample sizes are sufficient. For qualitative data, please indicate whether data saturation was considered, and what criteria were used to decide that no further sampling was needed.* |
| Data collection | *Provide details about the data collection procedure, including the instruments or devices used to record the data (e.g. pen and paper, computer, eye tracker, video or audio equipment) whether anyone was present besides the participant(s) and the researcher, and whether the researcher was blind to experimental condition and/or the study hypothesis during data collection.* |
| Timing | *Indicate the start and stop dates of data collection. If there is a gap between collection periods, state the dates for each sample cohort.* |
| Data exclusions | *If no data were excluded from the analyses, state so OR if data were excluded, provide the exact number of exclusions and the rationale behind them, indicating whether exclusion criteria were pre-established.* |
| Non-participation | *State how many participants dropped out/declined participation and the reason(s) given OR provide response rate OR state that no participants dropped out/declined participation.* |
| Randomization | *If participants were not allocated into experimental groups, state so OR describe how participants were allocated to groups, and if allocation was not random, describe how covariates were controlled.* |

# Ecological, evolutionary & environmental sciences study design

All studies must disclose on these points even when the disclosure is negative.

| | |
|---|---|
| Study description | *We conducted a whole-ecosystem mercury loading and abatement study (Mercury Experiment To Assess Atmospheric Loading In Canada And the United States; METAALICUS) to directly examine how quickly methylmercury concentrations in fish respond to reductions in inorganic mercury pollution. We added environmentally-relevant amounts of isotope-enriched inorganic mercury to a boreal lake and its catchment for 7 years, then ceased all isotopic inorganic mercury additions to simulate a reduction in mercury* |

loading. The lake, upland, and wetland areas of the catchment each received different isotopes, which we quantified (in addition to ambient mercury) in water, sediments and biota of the study lake during the addition phase and for 8 years following cessation of loading. We focus our analyses of fish methylmercury concentrations on three dominant species - yellow perch, northern pike, and lake whitefish - each important in commercial, subsistence and recreational fisheries. Temporal changes in ambient methylmercury concentrations were also monitored for yellow perch and northern pike in a nearby reference lake, consistent with standard whole-ecosystem study designs.

**Research sample**

Research samples were taken to quantify how different compartments of the lake responded to changes in inorganic mercury loading, and in turn how these compartments influenced fish methylmercury concentrations. Samples were taken during the open-water season and represent annual time-integrated estimates of isotopic and ambient methylmercury concentrations in water, the upper 2 cm of lake sediments, and the dominant invertebrate prey for fish (zooplankton, Chaoborus and chironomids). The dominant fish species in the lake were collected each autumn. We monitored young age classes of small-bodied fishes (yellow perch and blacknose shiner) and older age classes of large-bodied fish species (northern pike and lake whitefish).

**Sampling strategy**

Sampling in a small lake required careful consideration of potential disturbance to the lake (i.e., benthic and sediment sampling) and fish populations. We conducted large-scale pilot mesocosm studies prior to the whole-ecosystem experiment to inform sampling design and sample size. For small fish species, we collected only the youngest age classes, which are typically the most abundant, to avoid depleting these populations. We developed a non-destructive biopsy technique for large fish so that they could be sampled for methylmercury and returned to the lake. Many of the large fish were biopsied multiple times over the 15 year study which informed our understanding of methylmercury uptake and loss in individual fish. We collected samples of yellow perch and northern pike from a nearby reference lake to monitor annual changes in methylmercury concentrations of an undisturbed system.

**Data collection**

Data were collected over a period of 15 years (2001-2015) by Principal Investigators, graduate students, biologists/technicians, and research assistants. Data collection methods for each sample type are described in detail in the Methods. Briefly, water samples were collected by pump and filtered through in-line cartridges (tubing and filter apparatus were acid-cleaned Teflon). Sediment cores were collected by hand by divers and through use of a box corer. Zooplankton samples were collected by vertical tows of plankton nets through the water column. Benthic invertebrate samples were collected using an Ekman dredge. Fish samples were collected by trap net, gill net, pole seine net, and angling. All samples were processed using clean techniques for trace metals using stainless steel or Teflon tools. Mercury concentrations were quantified using multi-collector inductively-coupled plasma mass spectrometry (ICP-MS). Methylmercury was analysed in all samples, except for fish muscle tissue, where total mercury was quantified. For a subset of small-bodied fish we determined that most (>90%) of the isotopic and ambient mercury is methylmercury. Samples of certified reference materials were subjected to the same procedures; measured mercury concentrations in the reference materials were not statistically different from certified values.

**Timing and spatial scale**

Annual sampling frequency varied by compartment (e.g., bi-weekly or monthly for water, zooplankton; annually for fish) over the 15 year period (2001-2015). Sampling for water, sediments, zooplankton and invertebrates occurred during the most productive time of year (open-water season), at a frequency consistent with other whole-ecosystem studies. We consistently collected small- and large-bodied fish species annually from Lake 658 in the autumn months (September-November) as this represents the end of the growing season for north temperate fish species. All samples presented in this study were collected from within the experimental lake (8.4 ha surface area), and from a nearby reference lake.

**Data exclusions**

A year class failure of yellow perch resulted in a single young-of-year (YOY) collected in 2008 and no age 1+ fish in 2009. Data for the single YOY perch (n=1) captured in 2008 is not presented. A single northern pike biopsy was removed from all analyses due to unreliable mercury data.

**Reproducibility**

Results of whole ecosystem-scale experiments are rarely exactly reproducible. Results depend on a suite of ecosystem-specific conditions (e.g. for mercury, amount of associated wetland area or hypolimnetic anoxia could influence results), in addition to abiotic factors (e.g. temperature and precipitation) that can vary from year to year. Similar findings from smaller-scale pilot studies at the ELA (in different uplands, wetlands, and lakes) indicate that the findings from the METAALICUS project are generally reproducible.

**Randomization**

Randomization was not required. Fish samples were collected annually in autumn months from throughout the lake and were grouped in our analyses according to year. Water, sediment, invertebrate, and small-bodied fish (when considered as prey for northern pike) samples collected throughout each year were averaged to represent the mean methylmercury concentration in that compartment each year, as described in the Methods.

**Blinding**

Blinding was not necessary for this study because fish captures were random as was the selection of a subset of small fish for mercury concentration analyses.

Did the study involve field work?  ☒ Yes  ☐ No

# Field work, collection and transport

**Field conditions**

Field data collection occurred over a period of 15 years under a variety of field conditions throughout the open water season. The conditions did not influence data or sample collection methods. Annual air temperature from May through October at the ELA averaged 13.9 °C (range: 12.2-15.0 °C) with a range of observed temperatures between -10 °C and 35.5 °C. Cumulative annual precipitation averaged 537 mm (range: 389-734 mm) during the study. The application of enriched stable mercury isotopes to upland and wetland areas was achieved through crop-duster airplane and was undertaken each spring (if possible) just prior to a light rainfall event to simulate wet-deposition of mercury.

**Location**

Lake 658 (lake and watershed) is located in northwestern Ontario, Canada (49° 43' 95" N, 93° 44' 20" W; elevation 371 m). Upland catchment is 41.2 ha, wetland is 1.7 ha, and lake surface is 8.4 ha. Lake 658 is a double-basin lake with a maximum depth of 13 m. The reference lake (Lake 240; 44 ha) is located ~9 km due south of Lake 658.

**Access & import/export**

All research was conducted at the Experimental Lakes Area (ELA), a remote and pristine region of northwestern Ontario, Canada, where 58 lakes and their watersheds have been set aside for research on anthropogenic impacts to freshwater ecosystems. Lake 658

| Disturbance | and the reference lake (Lake 240) are designated ELA research lakes. From 1968 to 2014, ELA was a Federal Government facility. In 2014, ownership of ELA was transferred to the International Institute for Sustainable Development (now IISD-ELA). All research presented here was approved by provincial and federal authorities (see below). An annual License to Collect Fish for Scientific Purposes was awarded each year as required by the provincial government. |
|---|---|

*Whole-ecosystem research examining the effects of various human impacts on the environment has been ongoing at ELA since its inception in 1968. The METAALICUS project, as with all others conducted at ELA, was reviewed and approved by the ELA Research Advisory Board (Fisheries and Oceans Canada), Ontario Ministry of Natural Resources and Forestry, Ontario Ministry of the Environment and Climate Change, and Ontario Parks. The many layers of review and approval are in place to ensure that only important, legitimate, high quality science projects are undertaken at ELA. These agreements also ensure that lakes are returned to their natural state following manipulation.*

*In this project, the small amount of mercury added to the lake and watershed over 7 years (~1 teaspoon) did not pose a human health threat. Lake 658 itself is closed to recreational fishing and bait harvest, such that none of the fish from the lake were handled or removed by the public. A fence was installed at the outflow to prevent the the movement of fish between the downstream lake and Lake 658 for the duration of the study (as required by OMNRF). In accordance with monitoring requirements, a small number of fish in the downstream lake were collected annually.*

# Reporting for specific materials, systems and methods

We require information from authors about some types of materials, experimental systems and methods used in many studies. Here, indicate whether each material, system or method listed is relevant to your study. If you are not sure if a list item applies to your research, read the appropriate section before selecting a response.

## Materials & experimental systems

| n/a | Involved in the study |
|---|---|
| ☒ | ☐ Antibodies |
| ☒ | ☐ Eukaryotic cell lines |
| ☒ | ☐ Palaeontology and archaeology |
| ☐ | ☒ Animals and other organisms |
| ☒ | ☐ Human research participants |
| ☒ | ☐ Clinical data |
| ☒ | ☐ Dual use research of concern |

## Methods

| n/a | Involved in the study |
|---|---|
| ☒ | ☐ ChIP-seq |
| ☒ | ☐ Flow cytometry |
| ☒ | ☐ MRI-based neuroimaging |

## Antibodies

| Antibodies used | *Describe all antibodies used in the study; as applicable, provide supplier name, catalog number, clone name, and lot number.* |
|---|---|
| Validation | *Describe the validation of each primary antibody for the species and application, noting any validation statements on the manufacturer's website, relevant citations, antibody profiles in online databases, or data provided in the manuscript.* |

## Eukaryotic cell lines

Policy information about cell lines

| Cell line source(s) | *State the source of each cell line used.* |
|---|---|
| Authentication | *Describe the authentication procedures for each cell line used OR declare that none of the cell lines used were authenticated.* |
| Mycoplasma contamination | *Confirm that all cell lines tested negative for mycoplasma contamination OR describe the results of the testing for mycoplasma contamination OR declare that the cell lines were not tested for mycoplasma contamination.* |
| Commonly misidentified lines (See ICLAC register) | *Name any commonly misidentified cell lines used in the study and provide a rationale for their use.* |

## Palaeontology and Archaeology

| Specimen provenance | *Provide provenance information for specimens and describe permits that were obtained for the work (including the name of the issuing authority, the date of issue, and any identifying information). Permits should encompass collection and, where applicable, export.* |
|---|---|
| Specimen deposition | *Indicate where the specimens have been deposited to permit free access by other researchers.* |
| Dating methods | *If new dates are provided, describe how they were obtained (e.g. collection, storage, sample pretreatment and measurement), where they were obtained (i.e. lab name), the calibration program and the protocol for quality assurance OR state that no new dates are provided.* |

☐ Tick this box to confirm that the raw and calibrated dates are available in the paper or in Supplementary Information.

Ethics oversight | *Identify the organization(s) that approved or provided guidance on the study protocol, OR state that no ethical approval or guidance was required and explain why not.*

Note that full information on the approval of the study protocol must also be provided in the manuscript.

# Animals and other organisms

Policy information about studies involving animals; ARRIVE guidelines recommended for reporting animal research

Laboratory animals | *The study did not involve laboratory animals.*

Wild animals | *All fish were wild strain. In total, 285 yellow perch (Perca flavescens, 145 young-of-year and 140 age-1) and 138 blacknose shiner (Notropis heterolepis, age-1) were captured by pole seine net, small mesh gill net (20 minute set), and hoop net. These fish were euthanized in an overdose bath of tricaine methanesulfonate (0.25 g/L). Most small fish had not reached sexual maturity.*

*Northern pike (Esox lucius, aged at 2-12 y), lake whitefish (Coregonus clupeaformis, aged at 3-38 y), and white sucker (Catostomus commersonii, not aged) were captured by angling or multi-mesh gill nets (20-30 min set). Fish were anesthetized with tricaine methanesulfonate (0.06 g/L), basic biological information was collected, fish were tagged with a Passive Integrated Transponder (PIT) tag, and a small biopsy of dorsal muscle was collected using a dermal punch which was then sealed with veterinary tissue adhesive (VetBond). Fish were allowed to recover in a tub of fresh lake water (for ~15 min) before being released back into the lake. In total, 690 biopsy muscle samples were collected from 390 fish (238 northern pike, 114 lake whitefish, 38 white sucker) from 2001-2015 in Lake 658; 149 fish (90 northern pike, 38 lake whitefish, 21 white sucker) were biopsied more than once during the 15 year study (2 to 6 biopsies per individual). Sex data were not available for most fish as they were mostly captured outside of their spawning season.*

*Collection of zooplankton and benthic macroinvertebrates is not regulated or licenced in Ontario. Zooplankton samples were collected by vertical tows of plankton nets through the water column. Benthic invertebrate samples were collected using an Ekman dredge.*

Field-collected samples | *The study did not house samples collected from the field in the lab.*

Ethics oversight | *All work with vertebrate animals was approved by Animal Care Committees (ACC) through the Canadian Council on Animal Care (Freshwater Institute ACC for Fisheries & Oceans Canada, 2001-2013; University of Manitoba ACC for IISD-ELA, 2014-2015). Licenses to Collect Fish for Scientific Purposes were granted annually by the Ontario Ministry of Natural Resources and Forestry.*

Note that full information on the approval of the study protocol must also be provided in the manuscript.

# Human research participants

Policy information about studies involving human research participants

Population characteristics | *Describe the covariate-relevant population characteristics of the human research participants (e.g. age, gender, genotypic information, past and current diagnosis and treatment categories). If you filled out the behavioural & social sciences study design questions and have nothing to add here, write "See above."*

Recruitment | *Describe how participants were recruited. Outline any potential self-selection bias or other biases that may be present and how these are likely to impact results.*

Ethics oversight | *Identify the organization(s) that approved the study protocol.*

Note that full information on the approval of the study protocol must also be provided in the manuscript.

# Clinical data

Policy information about clinical studies

All manuscripts should comply with the ICMJE guidelines for publication of clinical research and a completed CONSORT checklist must be included with all submissions.

Clinical trial registration | *Provide the trial registration number from ClinicalTrials.gov or an equivalent agency.*

Study protocol | *Note where the full trial protocol can be accessed OR if not available, explain why.*

Data collection | *Describe the settings and locales of data collection, noting the time periods of recruitment and data collection.*

Outcomes | *Describe how you pre-defined primary and secondary outcome measures and how you assessed these measures.*

# Dual use research of concern

Policy information about dual use research of concern

## Hazards

Could the accidental, deliberate or reckless misuse of agents or technologies generated in the work, or the application of information presented in the manuscript, pose a threat to:

No | Yes

☐ | ☐ Public health

☐ | ☐ National security

☐ | ☐ Crops and/or livestock

☐ | ☐ Ecosystems

☐ | ☐ Any other significant area

## Experiments of concern

Does the work involve any of these experiments of concern:

No | Yes

☐ | ☐ Demonstrate how to render a vaccine ineffective

☐ | ☐ Confer resistance to therapeutically useful antibiotics or antiviral agents

☐ | ☐ Enhance the virulence of a pathogen or render a nonpathogen virulent

☐ | ☐ Increase transmissibility of a pathogen

☐ | ☐ Alter the host range of a pathogen

☐ | ☐ Enable evasion of diagnostic/detection modalities

☐ | ☐ Enable the weaponization of a biological agent or toxin

☐ | ☐ Any other potentially harmful combination of experiments and agents

# ChIP-seq

## Data deposition

☐ Confirm that both raw and final processed data have been deposited in a public database such as GEO.

☐ Confirm that you have deposited or provided access to graph files (e.g. BED files) for the called peaks.

Data access links
*May remain private before publication.*
> *For "Initial submission" or "Revised version" documents, provide reviewer access links. For your "Final submission" document, provide a link to the deposited data.*

Files in database submission
> *Provide a list of all files available in the database submission.*

Genome browser session
(e.g. UCSC)
> *Provide a link to an anonymized genome browser session for "Initial submission" and "Revised version" documents only, to enable peer review. Write "no longer applicable" for "Final submission" documents.*

## Methodology

Replicates
> *Describe the experimental replicates, specifying number, type and replicate agreement.*

Sequencing depth
> *Describe the sequencing depth for each experiment, providing the total number of reads, uniquely mapped reads, length of reads and whether they were paired- or single-end.*

Antibodies
> *Describe the antibodies used for the ChIP-seq experiments; as applicable, provide supplier name, catalog number, clone name, and lot number.*

Peak calling parameters
> *Specify the command line program and parameters used for read mapping and peak calling, including the ChIP, control and index files used.*

Data quality
> *Describe the methods used to ensure data quality in full detail, including how many peaks are at FDR 5% and above 5-fold enrichment.*

Software
> *Describe the software used to collect and analyze the ChIP-seq data. For custom code that has been deposited into a community repository, provide accession details.*

# Flow Cytometry

## Plots

Confirm that:

- [ ] The axis labels state the marker and fluorochrome used (e.g. CD4-FITC).
- [ ] The axis scales are clearly visible. Include numbers along axes only for bottom left plot of group (a 'group' is an analysis of identical markers).
- [ ] All plots are contour plots with outliers or pseudocolor plots.
- [ ] A numerical value for number of cells or percentage (with statistics) is provided.

## Methodology

| | |
|---|---|
| Sample preparation | *Describe the sample preparation, detailing the biological source of the cells and any tissue processing steps used.* |
| Instrument | *Identify the instrument used for data collection, specifying make and model number.* |
| Software | *Describe the software used to collect and analyze the flow cytometry data. For custom code that has been deposited into a community repository, provide accession details.* |
| Cell population abundance | *Describe the abundance of the relevant cell populations within post-sort fractions, providing details on the purity of the samples and how it was determined.* |
| Gating strategy | *Describe the gating strategy used for all relevant experiments, specifying the preliminary FSC/SSC gates of the starting cell population, indicating where boundaries between "positive" and "negative" staining cell populations are defined.* |

- [ ] Tick this box to confirm that a figure exemplifying the gating strategy is provided in the Supplementary Information.

# Magnetic resonance imaging

## Experimental design

| | |
|---|---|
| Design type | *Indicate task or resting state; event-related or block design.* |
| Design specifications | *Specify the number of blocks, trials or experimental units per session and/or subject, and specify the length of each trial or block (if trials are blocked) and interval between trials.* |
| Behavioral performance measures | *State number and/or type of variables recorded (e.g. correct button press, response time) and what statistics were used to establish that the subjects were performing the task as expected (e.g. mean, range, and/or standard deviation across subjects).* |

## Acquisition

| | |
|---|---|
| Imaging type(s) | *Specify: functional, structural, diffusion, perfusion.* |
| Field strength | *Specify in Tesla* |
| Sequence & imaging parameters | *Specify the pulse sequence type (gradient echo, spin echo, etc.), imaging type (EPI, spiral, etc.), field of view, matrix size, slice thickness, orientation and TE/TR/flip angle.* |
| Area of acquisition | *State whether a whole brain scan was used OR define the area of acquisition, describing how the region was determined.* |

Diffusion MRI  [ ] Used  [ ] Not used

## Preprocessing

| | |
|---|---|
| Preprocessing software | *Provide detail on software version and revision number and on specific parameters (model/functions, brain extraction, segmentation, smoothing kernel size, etc.).* |
| Normalization | *If data were normalized/standardized, describe the approach(es): specify linear or non-linear and define image types used for transformation OR indicate that data were not normalized and explain rationale for lack of normalization.* |
| Normalization template | *Describe the template used for normalization/transformation, specifying subject space or group standardized space (e.g. original Talairach, MNI305, ICBM152) OR indicate that the data were not normalized.* |
| Noise and artifact removal | *Describe your procedure(s) for artifact and structured noise removal, specifying motion parameters, tissue signals and physiological signals (heart rate, respiration).* |

Volume censoring | *Define your software and/or method and criteria for volume censoring, and state the extent of such censoring.*

## Statistical modeling & inference

Model type and settings | *Specify type (mass univariate, multivariate, RSA, predictive, etc.) and describe essential details of the model at the first and second levels (e.g. fixed, random or mixed effects; drift or auto-correlation).*

Effect(s) tested | *Define precise effect in terms of the task or stimulus conditions instead of psychological concepts and indicate whether ANOVA or factorial designs were used.*

Specify type of analysis: ☐ Whole brain  ☐ ROI-based  ☐ Both

Statistic type for inference
(See Eklund et al. 2016) | *Specify voxel-wise or cluster-wise and report all relevant parameters for cluster-wise methods.*

Correction | *Describe the type of correction and how it is obtained for multiple comparisons (e.g. FWE, FDR, permutation or Monte Carlo).*

## Models & analysis

n/a | Involved in the study
☐ | ☐ Functional and/or effective connectivity
☐ | ☐ Graph analysis
☐ | ☐ Multivariate modeling or predictive analysis

Functional and/or effective connectivity | *Report the measures of dependence used and the model details (e.g. Pearson correlation, partial correlation, mutual information).*

Graph analysis | *Report the dependent variable and connectivity measure, specifying weighted graph or binarized graph, subject- or group-level, and the global and/or node summaries used (e.g. clustering coefficient, efficiency, etc.).*

Multivariate modeling and predictive analysis | *Specify independent variables, features extraction and dimension reduction, model, training and evaluation metrics.*

