## [Peer Review File · Nature]

Manuscript Title: Experimental evidence for recovery of mercury-contaminated fish populations

Reviewer Comments & Author Rebuttals

Reviewer Reports on the Initial Version:

Referee #1 (Remarks to the Author):

General comments:

This paper presents the results of a unique and comprehensive whole ecosystem experiment that simulates atmospheric deposition of mercury to freshwater ecosystems. The results provide mechanistic evidence for a two phased response in biota based on an unprecedented 15 years of monitoring data. The study of the food web is detailed and comprehensive.

Results show a rapid initial decline in fish mercury, followed by a longer second phase response dictated by the watershed. Eight years after the cessation of the isotope additions, only a small fraction of the original atmospheric isotope spike remains in the fish. These results reinforce the potential benefits of reducing atmospheric mercury deposition on a broad scale through regulatory action. In my opinion, this simple presentation of results enabled by this very complicated experiment is badly needed to inform and incentivize global action. My only hesitation about this manuscript is that the scientific conclusions from an earlier version of the work (Harris et al., 2007) are essentially the same. The conclusions are illustrated in greater detail in this work with additional fish species and data from the recovery phase. I am worried that many members of the community have forgotten about this important work from 2007 so I encourage publication of this paper.

Specific comments:

Generally, the paper is clear and well written. I was a bit confused about what is going on with the individual pike in Figure 3 and how those data were converted to the population response? Some more discussion of patterns in ambient Hg (Figure 2) compared to the spike would be helpful – perhaps in the SI. It would also be helpful to have information on the mass budget for the system instead of just concentrations for Figure 1.

Referee #2 (Remarks to the Author):

Overall, I find that the results from this work deserve to be published/summarized in Nature (many other publications exist on this METAALICUS experiment). It does have far reaching implications (substantial recovery of food web in less than 10 years and some amount of spike Hg still sticking around), and it also broad audience appeal. Generally, the paper is well written (if a little dry - more active voice would help). I only have a couple of major comments and some minor comments below. One major comment is that in both in the abstract and intro, the name of the program is not included along with no context for the program/experiment. I think this is an odd omission, and that it would be helpful to the broad audience to include one to two sentences on how special/unique this program was in the context of Hg research. My other is that more context about how generalizable the results are is needed especially in the context of the upland and wetland data (and the implications of not seeing those spikes in the food web). I guess I feel like under the scenario of the study, impacts from terrestrial and wetland Hg is not really being studied in the way they would most likely impact downstream aquatic food webs (i.e., through increased erosion/input due to some sort of land use change). Thus, I think the authors need to be careful in there discussion of the Hg that is still being stored in those compartments. If they go back and

deforest the upland or somehow disrupt the wetland (change it/modify it like humans do), then they could assess the impact of that Hg (that would be a cool experiment).

Line by line comments:

Line 51: I suspect that biomagnification is more important in increasing Hg concentrations up the food chain, but would not bioaccumulation be more encompassing of other concentrating pathways?

Line 55: There is a lot of passive voice in this opening. Suggest trying to remove some of it such as the "has hampered" to just "hampers" to throw in some active voice for the reader.

Line 80: Suggest removing passive voice: Starting in 2001, we conducted....

Line 105: I feel a sentence here would be nice to tell the reader that these sources can be important in other instances – such as through increased erosion due to land use change or some thing like that. And then just say that this study did not assess that since the upland and wetland systems stayed intact.

Line 119: comma unnecessary after 60%

Line 189: I feel that the definition of rapid (<10 years) should be introduced earlier as I was going to comment that this was undefined in much of the text before this.

Figure 1: This is a hard graph for an outside reader to figure out what line goes with what. I would prefer sed and water being one type of symbol and then other types of samples other symbols. Right now it is unwieldy for someone not familiar with the data.

Line 329: I wish this definition of biomagnification was also in text

Figure 2: The pictures are not that helpful and I would prefer the name of the fish on the individual graphs. Plus for d-f, the names of predator fish and their prey would also be useful.

Referee #3 (Remarks to the Author):

Blanchfield and coworkers present a follow-up study of their unique Experimental Lake Area (ELA) study started 20y ago. The authors initially observed that the enriched inorganic Hg spike added from 2001-2007 to a small boreal lake rapidly infiltrated the foodweb in the methyl-Hg form. Two additional spikes added to the upland catchment and wetland did not show up in foodweb MeHg (Harris et al., 2007). In this new study they show that since 2009, apex fish spike methyl-Hg levels have dropped about 50% over the 8 subsequent years. The authors argue this provides evidence for a potential rapid recovery of fisheries with health benefits for fish consumers, if the Minamata Convention on Hg is rigorously translated into lower global Hg emissions to the atmosphere.

I have always appreciated the audacity of the ELA Hg studies, and the knowledge gained on Hg cycling over the years has been substantial, and will continue to grow in the decades to come as the upland and wetland spikes make it to the lake. The question addressed is of great importance, and the claims made by such a large number of Hg specialists are important, and potentially encouraging for environmental policy makers, and therefore of broad importance. Yet that doesn't mean that the findings and wider impacts of the study are as robust as the well-written MS portrays. I see multiple issues of representativeness of the results and implications stated, and important limitations of the spike approach that need clarification:

1. Regardless of the size of an isotopic or elemental spiking experiment, i.e. whole lake or lab erlenmeyer, the caveats remain the same: a spike must ideally be introduced in the same molecular form and concentration as its natural equivalent, if the spike is to show similar reactivity.

Since these aspects are not discussed in the MS, I have examined the concentration of inorganic Hg spike released into lake surface water, based on the SI: Lake area=8.4 ha; 22 ug Hg spike/m²/y; 4x20L carboys with lake water acidified to pH4. This means the carboys contained $8.4 \times 10000 \times 1 \times 22 / 4 / 20 = 23 \text{mg/L}$ of spike Hg in each carboy. This concentration is approximately 10,000,000 higher than the ambient lake Hg of 1.7 ng/L (Harris et al., 2007). The natural lake pH appears to be 6.5. It implies that the spike Hg speciation in the 20L carboy most likely remained in the labile Hg(NO₃)₂ form, and therefore bioavailability was not representative of ambient conditions, where inorganic Hg is strongly bound to S groups on DOM. Have the authors addressed these issues in a previous paper? I understand that the final diluted concentration of the spike is close to natural levels of 1.7 ng/L, but the labile Hg speciation during the weeks necessary for dilution may be critical in controlling spike methylation potential. Spiking experiments have come a long way since 2001: For ex. recent work using Hg spikes dosed as labile, NOM bound, and S-bound Hg forms show order of magnitude differences in methylation rates (Jonsson et al., 2014).

2. A related question I have, concerns what MeHg source in the boreal lake watershed dominates fish MeHg levels? Watershed MeHg? Water column MeHg or sediment and anoxic layer MeHg? Over the last decade, Hg methylation has been evidenced in oxic water columns of many waterbodies, including the ocean. Was oxic water column methylation following spike addition studied during the experiment? Other studies have found watershed MeHg to be a dominant contributor to lake MeHg (Bravo et al., 2017). Finally, what is the source of inorganic Hg in ambient fish MeHg? In the experiment and MS the lake spike represents Hg wet deposition to the lake. Twenty years ago we indeed considered that lakes were mostly supplied with Hg via wet deposition, but subsequent work has pointed out the important contribution of Hg in litterfall and throughfall to soils in the ELA watershed (St. Louis et al., 2019) which runs off into the lake.

3. In one of their key papers on the ELA spike experiment following the spike loading phase the authors concluded 'However, a full response will be delayed by the gradual export of mercury stored in watersheds. The rate of response will vary among lakes depending on the relative surface areas of water and watershed.' And 'We expect that, in the long term, the full effect of the increased mercury loading will be much larger than the 30–40% increase in methylmercury in the biota that was evident after only 3 years.' (Harris et al., 2007). Why is this finding is no longer discussed in the present MS? An interesting feature of the cited long Hg residence time in watersheds is visible in N-American lake sediment cores, where Hg sedimentation rates over the past 40 years is constant (Fitzgerald et al., 2005; Engstrom et al., 2014), while Hg emission, atmospheric Hg concentrations and Hg wet deposition have dropped by a factor of 2. In other words, watershed contributions to lake Hg (and MeHg) appear to be dominant, and recovery is rather slow.

4. This is a spike study on one single lake; foodweb MeHg dynamics depend on the full biogeochemistry of the watershed and lake: climate, C-cycling and OM quality; soil and lake pH; biodiversity etc etc. It is common to address important questions such as ecosystem recovery or Hg re-emissions based on the review of dozens to hundreds of studies. The ELA study alone cannot fulfill that criteria unfortunately.

As I mentioned above I tremendously respect the time and energy invested in the ELA project; it was designed as best as one could at that time. I am however not convinced, based on the approach and results presented, that the swift freshwater ecosystem recovery is a robust and globally representative observation.

Detailed comments:

Title; suggest to formulate as '..recovery of mercury-contaminated fish in boreal lakes'

L106 'Delivery of lake spike inorganic Hg to sites of methylation, including sediments and anoxic bottom waters,..'

Was there evidence of methylation in the oxic lake water column? Is the fraction of foodweb MeHg originating from catchment runoff, water column methylation, and sediment methylation known? Scandinavian work on boreal lakes suggest a dominant role for catchment runoff (ex: (Bravo et al., 2017)

L96 'The study catchment is located in a remote region of Canada that experiences low levels of atmospheric pollution, such that our experimental loading rate increased wet Hg deposition ~5-fold (from ~3.6 to ~19 $\mu\text{g m}^{-2} \text{y}^{-1}$), similar to more polluted regions of the world²².' Remote lake sediment cores across N-America generally show 3x enrichment since 1850, so I'm not sure 'low level of pollution' is appropriate.

L98 'Most of the Hg added to the upland and wetland areas surrounding Lake 658 either remained bound to vegetation and soils or evaded back to the atmosphere^{23,24}. The Hg applied to the upland catchment accounted for only a small fraction (<1%) of all Hg in runoff to the lake¹⁹ and consequently contributed little (maximum <5%) to overall fish MeHg concentrations throughout the study (Extended Data Fig. 1).'

I am not surprised by the slow spike Hg release from the catchment, but I am puzzled why ambient inorganic Hg and MeHg in catchment runoff are subsequently dismissed as an important source to lake food web MeHg. Visibly, 99% of inorganic Hg in runoff from the catchment, typically bound to DOC in boreal environment, is sustained and transiting into the lake where it is available for methylation. Adding a spike upland leads to a strong disequilibrium, and it may take decades before that spike shows up downstream; this doesn't mean that upland Hg is not an important source.

Nature reviewer checklist: Raw data for Figures and discussion is not provided; Data statistics are however provided in the SI and are appropriate. Error bars are missing in most Figures or legends. Previous literature is correctly cited. MS is clear and accessible.

References cited

- Bravo, A. G., Bouchet, S., Tolu, J., Björn, E., Mateos-Rivera, A., and Bertilsson, S.: Molecular composition of organic matter controls methylmercury formation in boreal lakes, *Nat. Commun.*, 8, 14255, <https://doi.org/10.1038/ncomms14255>, 2017.
- Engstrom, D. R., Fitzgerald, W. F., Cooke, C. A., Lamborg, C. H., Drevnick, P. E., Swain, E. B., Balogh, S. J., and Balcom, P. H.: Atmospheric Hg Emissions from Preindustrial Gold and Silver Extraction in the Americas: A Reevaluation from Lake-Sediment Archives, *Environ. Sci. Technol.*, 48, 6533–6543, <https://doi.org/10.1021/es405558e>, 2014.
- Fitzgerald, W. F., Engstrom, D. R., Lamborg, C. H., Tseng, C.-M., Balcom, P. H., and Hammerschmidt, C. R.: Modern and Historic Atmospheric Mercury Fluxes in Northern Alaska: Global Sources and Arctic Depletion, *Environ. Sci. Technol.*, 39, 557–568, <https://doi.org/10.1021/es049128x>, 2005.
- Harris, R. C., Rudd, J. W. M., Amyot, M., Babiarz, C. L., Beaty, K. G., Blanchfield, P. J., Bodaly, R. A., Branfireun, B. A., Gilmour, C. C., Graydon, J. A., Heyes, A., Hintelmann, H., Hurley, J. P., Kelly, C. A., Krabbenhoft, D. P., Lindberg, S. E., Mason, R. P., Paterson, M. J., Podemski, C. L., Robinson, A., Sandilands, K. A., Southworth, G. R., Louis, V. L. S., and Tate, M. T.: Whole-ecosystem study shows rapid fish-mercury response to changes in mercury deposition, *Proc. Natl. Acad. Sci. U. S. A.*, 104, 16586–16591, <https://doi.org/10.1073/pnas.0704186104>, 2007.
- Jonsson, S., Skyllberg, U., Nilsson, M. B., Lundberg, E., Andersson, A., and Björn, E.: Differentiated availability of geochemical mercury pools controls methylmercury levels in estuarine sediment and biota, *Nat. Commun.*, 5, 4624, <https://doi.org/10.1038/ncomms5624>, 2014.
- St. Louis, V. L., Graydon, J. A., Lehnher, I., Amos, H. M., Sunderland, E. M., St. Pierre, K. A., Emmerton, C. A., Sandilands, K., Tate, M., Steffen, A., and Humphreys, E. R.: Atmospheric Concentrations and Wet/Dry Loadings of Mercury at the Remote Experimental Lakes Area, Northwestern Ontario, Canada, *Environ. Sci. Technol.*, 53, 8017–8026, <https://doi.org/10.1021/acs.est.9b01338>, 2019.

Author Rebuttals to Initial Comments:

Referee #1 (Remarks to the Author):

General comments:

This paper presents the results of a unique and comprehensive whole ecosystem experiment that simulates atmospheric deposition of mercury to freshwater ecosystems. The results provide mechanistic evidence for a two phased response in biota based on an unprecedented 15 years of monitoring data. The study of the food web is detailed and comprehensive.

Results show a rapid initial decline in fish mercury, followed by a longer second phase response dictated by the watershed. Eight years after the cessation of the isotope additions, only a small fraction of the original atmospheric isotope spike remains in the fish. These results reinforce the potential benefits of reducing atmospheric mercury deposition on a broad scale through regulatory action. In my opinion, this simple presentation of results enabled by this very complicated experiment is badly needed to inform and incentivize global action.

We thank the referee for their positive remarks on our study.

My only hesitation about this manuscript is that the scientific conclusions from an earlier version of the work (Harris et al., 2007) are essentially the same. The conclusions are illustrated in greater detail in this work with additional fish species and data from the recovery phase. I am worried that many members of the community have forgotten about this important work from 2007 so I encourage publication of this paper.

METAALICUS was a two-phase experiment that included a distinct Hg loading phase (of 6-7 years), immediately followed by a recovery phase once Hg loadings were stopped. The Harris et al. (2007) publication was often criticized because it only described the findings from the **initial three years** of the mercury *loading* phase and did not look at what happens during recovery from decreased loading. Also, as noted by the Referee, this earlier paper contained limited fish data (1 species) that did not include any of the large-bodied fish species that are often consumed by humans.

The ultimate goal of METAALICUS was to determine if *reducing* inputs of Hg to lakes and their watersheds would reduce the amount of MeHg in fish once they accumulated it. That important policy question could not be answered by Harris et al. (2007). The recovery phase of the experiment was critical to test if MeHg in fish would go down at all after we stopped loading, because we knew that the Hg additions that had gone to the sediments would remain there in the long term, and could potentially continue to contribute to ongoing MeHg production. Fish MeHg did decline, albeit more slowly to reductions in Hg pollution than predicted from the loading phase. This is stated more explicitly in this version (lines 202-213; Extended Data Fig. 3). Throughout the manuscript we specifically discuss the loading and recovery phases of the experiment separately and make a clear distinction between these phases in the presentation of our data (all figures and tables).

Specific comments:

Generally, the paper is clear and well written. I was a bit confused about what is going on

with the individual pike in Figure 3 and how those data were converted to the population response?

We have attempted to clarify any confusion regarding individual versus population-level recovery of pike. In this revised version, we have stated that the population data are comprised of all fish sampled in a given year, and therefore also include some of the individual fish we were able to follow through time.

Elimination of bioaccumulated MeHg in boreal fishes is quite slow (Madenjian et al. 2021). Therefore, the decline in MeHg concentrations in fish populations in the recovery phase of our experiment was largely from the recruitment of new (i.e., young) fish into the population and the loss of older fish. This was because new pike born into the lake after Hg loading ceased accumulated very little spike Hg. Following reductions in loading, as the older pike in the lake were replaced by younger pike, the average concentration of spike Hg in the population as a whole declined even though some of the older pike retained their Hg. In order to respond to the request for clarification by the Referee, the entire paragraph was edited (lines 181-193), as were the captions for Figure 3 (lines 359-368) and Extended Data Figure 2.

Some more discussion of patterns in ambient Hg (Figure 2) compared to the spike would be helpful – perhaps in the SI.

In the original version of the paper, we had stated that ambient Hg in fish remained relatively stable (or in some cases declined) in the METAALICUS lake and in our reference lake (Lake 240) throughout the 15-year study (Extended Data Table 1). In this revised version, we elaborate on this point by including a statement (lines 116-121) that the relatively stable ambient Hg concentrations in fish were largely influenced by the relatively stable watershed inputs of Hg, which was the dominant source of ambient inorganic Hg to both the METAALICUS lake (from Harris et al. 2007; see next comment) and the reference lake (Sellers et al. 2001; now included as reference 26). This more direct discussion of the ambient Hg results and the sources of this ambient Hg (i.e., watershed) also addresses questions asked by Referees #2 and #3 on this topic.

It would also be helpful to have information on the mass budget for the system instead of just concentrations for Figure 1.

Mass inputs of ambient and isotopic Hg were provided for the initial three years of loading in Harris et al. (2007). Full budgets (including mass outflows, volatilization, etc.) are beyond the scope of this paper and would require a lot of additional data. This paper is focused directly on what happened with the fish in response to increased and decreased Hg loadings.

Referee #2 (Remarks to the Author):

Overall, I find that the results from this work deserve to be published/summarized in Nature (many other publications exist on this METAALICUS experiment). It does have far reaching implications (substantial recovery of food web in less than 10 years and some amount of spike Hg still sticking around), and it also broad audience appeal. Generally, the paper is well

written (if a little dry - more active voice would help). I only have a couple of major comments and some minor comments below. One major comment is that in both in the abstract and intro, the name of the program is not included along with no context for the program/experiment. I think this is an odd omission, and that it would be helpful to the broad audience to include one to two sentences on how special/unique this program was in the context of Hg research.

These are excellent suggestions. We have edited the manuscript to remove passive voice and to generally improve the writing. In the revised version we now include the name of the project (METAALICUS) in the main body of the text (lines 79-80, 90, 95, 203, 218), and have added some additional information to highlight the uniqueness of this whole-ecosystem experiment (lines 80-89). We opted to not include the project name in the Abstract, where defining the acronym would consume too much of the 200 word limit.

My other is that more context about how generalizable the results are is needed especially in the context of the upland and wetland data (and the implications of not seeing those spikes in the food web). I guess I feel like under the scenario of the study, impacts from terrestrial and wetland Hg is not really being studied in the way they would most likely impact downstream aquatic food webs (i.e., through increased erosion/input due to some sort of land use change). Thus, I think the authors need to be careful in there discussion of the Hg that is still being stored in those compartments. If they go back and deforest the upland or somehow disrupt the wetland (change it/modify it like humans do), then they could assess the impact of that Hg (that would be a cool experiment).

The purpose of METAALICUS was to study the effects of changing mercury deposition, in the absence of other changes to the watershed and lake. However, we agree with the Referee's notion that disturbing the watershed in some way might accelerate the movement of spike Hg into the lake, and have included a statement to this effect (lines 103-105). For example, boreal forests burn every 50-100 years, which may result in a pulse of inorganic Hg into the lake. The extent to which our results can be generalized to other lakes is an important issue (also raised by Referee #3) that prompted further discussion of the spike Hg stored in the watershed, which we now address in a revised final paragraph to the manuscript (lines 214-229).

Line by line comments:

Line 51: I suspect that biomagnification is more important in increasing Hg concentrations up the food chain, but would not bioaccumulation be more encompassing of other concentrating pathways?

We are not entirely clear what the Referee means here. We do agree that bioaccumulation into organisms is certainly important (and this is mentioned on line 53 of the Abstract), but since MeHg increases with each trophic level, biomagnification is really the important process as it relates to human health, which is the point we are making here. It is why fish MeHg concentrations are often thousands to millions of times higher than those in water (Fig. 1b). We have not made any changes to the text.

Line 55: There is a lot of passive voice in this opening. Suggest trying to remove some of it such as the “has hampered” to just “hampers” to throw in some active voice for the reader.

We made this particular change, as well as edited the entire manuscript to remove passive voice and improve the writing.

Line 80: Suggest removing passive voice: Starting in 2001, we conducted....

We made this change.

Line 105: I feel a sentence here would be nice to tell the reader that these sources can be important in other instances – such as through increased erosion due to land use change or some thing like that. And then just say that this study did not assess that since the upland and wetland systems stayed intact.

We agree (see response to major comment above).

Line 119: comma unnecessary after 60%

Corrected.

Line 189: I feel that the definition of rapid (<10 years) should be introduced earlier as I was going to comment that this was undefined in much of the text before this.

The definition of rapid (<10 years) to describe the decline of isotopic MeHg has been moved earlier to its' first mention in the main text (line 148).

Figure 1: This is a hard graph for an outside reader to figure out what line goes with what. I would prefer seds and water being one type of symbol and then other types of samples other symbols. Right now it is unwieldy for someone not familiar with the data.

We agree that Figure 1 is somewhat challenging to interpret given the large amount of data. We do feel that the inclusion of all lake ecosystem components is necessary for the reader to interpret the response of the different fish species to changes in Hg load to the lake. With that said, we have now tried to improve the readability of the figure by using a new combination of colours and symbols.

Line 329: I wish this definition of biomagnification was also in text

The formula used to calculate MeHg biomagnification factors is provided in the caption of Figure 2 (line 352), and also in the Methods (lines 550-560). We do not feel it also needs to be included in the main text. We have edited this paragraph to improve the writing and

believe the present text more clearly conveys biomagnification as the increase in MeHg from prey to predator.

Figure 2: The pictures are not that helpful and I would prefer the name of the fish on the individual graphs. Plus for d-f, the names of predator fish and their prey would also be useful.

We feel that the pictures provide a visualization of the differences in prey and trophic structure for these three key fish species. We did try a version with the fish and prey shown and labelled on the figure itself, but we found this version overly cluttered. We have now revised the figure caption to include the names of the prey items in each of the panels for clarity (lines 352-355). The names of the prey items consumed by each fish species are also clearly stated in the main text (lines 134-138; 170-180) and in the Methods (lines 557-560). The pictures are labelled in Figure 1, and their inclusion in Figure 2 aids with the flow of ideas in the manuscript (temporal changes in concentration among biota and biomagnification); however, we are quite willing to go with the editor's preference in this instance.

Referee #3 (Remarks to the Author):

Blanchfield and coworkers present a follow-up study of their unique Experimental Lake Area (ELA) study started 20y ago. The authors initially observed that the enriched inorganic Hg spike added from 2001-2007 to a small boreal lake rapidly infiltrated the foodweb in the methyl-Hg form. Two additional spikes added to the upland catchment and wetland did not show up in foodweb MeHg (Harris et al., 2007).

We would like to respectfully point out that Harris et al. (2007) included data only from the first three years (2001-2003) of isotopic Hg loading to the METAALICUS watershed, not the entire loading period (2001-2007). It was published in 2007, so that is likely the source of confusion here.

In this new study they show that since 2009, apex fish spike methyl-Hg levels have dropped about 50% over the 8 subsequent years. The authors argue this provides evidence for a potential rapid recovery of fisheries with health benefits for fish consumers, if the Minamata Convention on Hg is rigorously translated into lower global Hg emissions to the atmosphere.

To clarify, the eight years over which we conducted the recovery phase started in 2008; the last spike went into the lake in October 2007 (see Table 12 in response below). Lake spike MeHg concentrations (and burdens) in the apex predator, northern pike, dropped by 50% within a span of 4-5 years (see Fig. 3). A 50% reduction in lake spike MeHg took eight years for lake whitefish. We do think that the above is clearly described in the text.

I have always appreciated the audacity of the ELA Hg studies, and the knowledge gained on Hg cycling over the years has been substantial, and will continue to grow in the decades to come as the upland and wetland spikes make it to the lake. The question addressed is of great importance, and the claims made by such a large number of Hg specialists are important, and potentially encouraging for environmental policy makers, and therefore of broad importance.

Yet that doesn't mean that the findings and wider impacts of the study are as robust as the well-written MS portrays. I see multiple issues of representativeness of the results and implications stated, and important limitations of the spike approach that need clarification:

Thank you for both your positive remarks and more general comments/concerns, which we address below.

1. Regardless of the size of an isotopic or elemental spiking experiment, i.e. whole lake or lab erlenmeyer, the caveats remain the same: a spike must ideally be introduced in the same molecular form and concentration as its natural equivalent, if the spike is to show similar reactivity.

Since these aspects are not discussed in the MS, I have examined the concentration of inorganic Hg spike released into lake surface water, based on the SI: Lake area=8.4 ha; 22 ug Hg spike/m²/y; 4x20L carboys with lake water acidified to pH4. This means the carboys contained $8.4 \times 10000 \times 1 \times 22 / 4 / 20 = 23 \text{ mg/L}$ of spike Hg in each carboy. This concentration is approximately 10,000,000 higher than the ambient lake Hg of 1.7 ng/L (Harris et al., 2007).

The calculations by the Referee are correct, although we did not add the Hg isotope to the lake all at once each summer, but every two weeks during the open-water season (for a total of 9 additions each year; also in the table below). This was done to better simulate the natural frequency of rain events onto the lake surface, and also to maintain the added Hg concentrations at trace levels in the receiving waters, and close to those normally found in ELA lakes. These details are included in the Methods (line 394-397) and are presented in Figure 1a of the manuscript. The exact amounts of inorganic spike Hg added to the upland, wetland and lake are provided in the Sandilands et al. (2008) report cited in the original manuscript (see below for details of lake spike additions).

Table 12 from Sandilands, K. A. et al. Application of enriched stable mercury isotopes to the Lake 658 watershed for the METAALICUS project, at the Experimental Lakes Area, northwestern Ontario, Canada, 2001–2007. Can. Tech. Rep. Fish. Aquat. Sci. 2813 (2008).

Table 12. Amount of Lake Spike Hg added to Lake 658 from 2001 to 2007.

2001		2002		2003		2004		2005		2006		2007	
Date	Lake Spike Hg added (mg)	Date	Lake Spike Hg added (mg)	Date	Lake Spike Hg added (mg)	Date	Lake Spike Hg added (mg)	Date	Lake Spike Hg added (mg)	Date	Lake Spike Hg added (mg)	Date	Lake Spike Hg added (mg)
19-Jun	206.13	4-Jun	208.47	10-Jun	208.18	15-Jun	210.94	14-Jun	207.33	13-Jun	179.92	12-Jun	219.33
3-Jul	206.33	18-Jun	207.90	24-Jun	207.83	29-Jun	210.94	28-Jun	206.33	27-Jun	179.92	26-Jun	218.22
17-Jul	205.25	2-Jul	207.30	9-Jul	207.75	13-Jul	210.94	12-Jul	205.88	11-Jul	179.92	12-Jul	217.24
1-Aug	204.95	16-Jul	206.67	22-Jul	207.53	27-Jul	210.94	26-Jul	206.06	25-Jul	179.92	24-Jul	217.10
15-Aug	204.70	30-Jul	206.92	6-Aug	208.38	10-Aug	210.94	9-Aug	206.18	9-Aug	179.92	7-Aug	217.26
28-Aug	206.28	13-Aug	206.80	20-Aug	208.23	24-Aug	210.94	23-Aug	206.51	22-Aug	179.92	22-Aug	217.79
11-Sep	206.13	27-Aug	206.37	3-Sep	208.28	7-Sep	210.94	7-Sep	206.66	6-Sep	179.92	5-Sep	216.79
25-Sep	206.35	10-Sep	206.54	16-Sep	208.18	21-Sep	210.94	21-Sep	206.66	20-Sep	179.92	18-Sep	217.45
9-Oct	206.30	24-Sep	199.08	30-Sep	207.91	5-Oct	210.94	4-Oct	206.39	3-Oct	179.92	2-Oct	218.77
TOTAL	1852.40		1856.05		1872.24		1898.50		1858.00		1619.30		1959.95
Area (m ²)	83924		83924		83924		83924		83924		83924		83924
Application Rate (ug/m²)	22.07		22.12		22.31		23.03		22.14		19.29		23.35

The natural lake pH appears to be 6.5. It implies that the spike Hg speciation in the 20L carboy most likely remained in the labile $\text{Hg}(\text{NO}_3)_2$ form, and therefore bioavailability was not representative of ambient conditions, where inorganic Hg is strongly bound to S groups on DOM. Have the authors addressed these issues in a previous paper?

The Referee is understandably concerned whether the bioavailability of spike Hg added to the surface waters of the METAALICUS lake influenced the findings of our study. We have not addressed this issue in a previous paper, although we have actually collected novel data as part of the METAALICUS experiment that specifically address this concern. These data have not been published previously, and we now include these findings in the Methods portion of our revised manuscript. As now explained in the Methods, it is impossible to add the Hg spike at concentrations found naturally in rain. Instead, the starting point of the experiment was to ensure that the spike was at trace concentrations (approximately 1 ng/L) and was behaving like ambient Hg very soon after its addition to the epilimnion. We investigated the success of this approach in two ways: 1) we demonstrated that the morning after the spike (i.e., within 12 hours) there was no bioavailable $\text{Hg}(\text{II})$, including any “free” $\text{Hg}(\text{II})$ from dissociation of $\text{Hg}(\text{NO}_3)_2$, in surface water as assayed by a bioreporter bacteria, which is the same as what we found for ambient $\text{Hg}(\text{II})$; and 2) we determined that the morning after the spike the % of total Hg that was dissolved gaseous Hg was the same for ambient and spike Hg. Taken together, these findings convinced us that we were as successful as possible in adding the spike to the lake in a way that it behaved like ambient Hg very soon after its additions. This information has been added to the Methods (lines 399-418), along with supporting literature (references 38 and 39).

Other experiments have clearly shown that Hg nanoparticles (often with sulfides and DOM), and Hg associated with solids, exhibit very slow exchange with the aqueous (and bioavailable) Hg pool. However, Hg salts, like the HgNO_3 used as lake spike in METAALICUS, are known to equilibrate rapidly with DOM. In our experience, and in the literature, $\text{Hg}(\text{II})$ and $\text{MeHg}(\text{II})$ salts come to equilibrium with DOM in just a few hours (c.f. Haitzer et al. 2002; Clarisse and Hintelmann 2006; Sanders et al. 2020).

We also note that $\text{Hg}(\text{II})$ in rain is in a low pH solution, complexed with smaller, more loosely-binding ligands (Vet et al. 2014) than in the receiving waters. So there is also an initial equilibration that occurs in natural rainfall events. However, from our measurements of bioavailability and DGM production soon after our spike additions, we expect this process to be rapid.

I understand that the final diluted concentration of the spike is close to natural levels of 1.7 ng/L, but the labile Hg speciation during the weeks necessary for dilution may be critical in controlling spike methylation potential. Spiking experiments have come a long way since 2001: For ex. recent work using Hg spikes dosed as labile, NOM bound, and S-bound Hg forms show order of magnitude differences in methylation rates (Jonsson et al., 2014).

As noted above, the spike was uniformly distributed in surface waters by the morning after addition, and our testing showed that it was behaving like ambient Hg. Hence, dilution of the Hg spikes into the epilimnion occurred in a matter of hours, not weeks. These findings are consistent with previous whole-lake studies at the ELA showing rapid horizontal distribution of newly-added isotopes, such that epilimnetic concentrations are homogenous the morning

after additions (e.g., Quay et al. 1979; Bower et al. 1987). We have made mention of this fact in the revised manuscript and now include one of the above references in the Methods (lines 397-398, reference 37).

In methylation assays in sediments in the laboratory, equilibration with NOM is important to simulate the fact that when Hg(II) arrives naturally into sediments, it has already been in the lake for some time and is already bound, at trace levels, to strong sulfhydryl moieties (Haitzer et al., 2002). In our whole-ecosystem experiment, the Hg(II) circulated and equilibrated with NOM in the surface water before arriving at sites of methylation in the sediments or anoxic (hypolimnetic) water column.

In advance of the whole-lake study, we conducted several pilot studies, including a 2-year isotopic Hg loading study using large (10-m diam.) in-lake mesocosms to specifically examine the question of single versus multiple Hg additions (Paterson et al. 2006). Based on these findings, we added the lake spike regularly throughout the season to mimic natural patterns of Hg delivery to the lake (as wet deposition), and to avoid overwhelming the system (see Table 12 above). We also added the Hg at dusk/night to minimize photo-reductive losses of this Hg to the atmosphere.

2. A related question I have, concerns what MeHg source in the boreal lake watershed dominates fish MeHg levels? Watershed MeHg? Water column MeHg or sediment and anoxic layer MeHg?

Over the last decade, Hg methylation has been evidenced in oxic water columns of many waterbodies, including the ocean. Was oxic water column methylation following spike addition studied during the experiment? Other studies have found watershed MeHg to be a dominant contributor to lake MeHg (Bravo et al., 2017).

In our earlier study, we showed that ambient and lake spike MeHg moved in temporal synchrony in water and various biota during the early loading phase (Fig. 4 of Harris et al. 2007). We concluded that this synchrony occurs because the same factors influencing the formation of lake spike MeHg *within the lake* are also influencing formation of ambient MeHg. If ambient MeHg produced in the watershed were an important source of ambient MeHg to fish in the study lake, this synchrony would not be apparent. Synchrony in ambient and lake spike MeHg therefore necessitates that the dominant source of ambient MeHg to fish is produced in the lake itself. We have included these earlier findings in the revised manuscript (lines 119-121).

Water column methylation did not occur in the oxic upper waters of METAALICUS lake (Eckley and Hintelmann, 2006). Rather, water column methylation was confined to just the very bottom depths of the lake that were anoxic at the time of sampling. This reference has now been included (reference 25), along with some revised text, when describing where Hg methylation occurred in the lake (lines 108-110).

The extent to which MeHg is formed in the watershed and transported to the lake (to eventually end up in fish) depends on the type of watershed and lake. As discussed above and now in the text of the paper (lines 119-121), the METAALICUS lake is dominated by in-

lake methylation. Also, as per the Referee's suggestion, we compared Lake 658 with the set of lakes examined in the Bravo et al. 2017 paper, where, according to their characterizations, some of their boreal lakes were dominated by catchment sources of MeHg and some by in-lake methylation. Measurements of methylation rate constants (k_m) in our study lake place it on the "in-lake" end of the spectrum of lakes that they studied. Our reference lake has also been demonstrated to be dominated by in-lake methylation (Sellers et al., 2001, reference 26).

Finally, what is the source of inorganic Hg in ambient fish MeHg? In the experiment and MS the lake spike represents Hg wet deposition to the lake. Twenty years ago we indeed considered that lakes were mostly supplied with Hg via wet deposition, but subsequent work has pointed out the important contribution of Hg in litterfall and throughfall to soils in the ELA watershed (St. Louis et al., 2019) which runs off into the lake.

Harris et al. (2007) showed that runoff from the watershed was the largest source of inorganic ambient Hg to the lake, and that the percent of ambient Hg as MeHg in watershed runoff was <1.5%, compared to 10-15% in the water column of the lake. This supports the idea that in-lake methylation is the dominant source of MeHg to the lake food web in this lake. As noted in the previous response, synchronous production of spike and ambient MeHg within the lake is evidence for *in-lake methylation* as the dominant source of ambient MeHg to the food web and to fish.

The Referee is correct that all spike applications were applied at environmentally-relevant concentrations to simulate wet Hg deposition rates in more polluted areas of North America and the world. This was approximately a 5X increase in wet deposition rates at the ELA. In the intervening years since we began this study, St. Louis et al. (2019) showed that, at the ELA, dry Hg loadings in throughfall and litterfall were 2.7 to 6.1 times greater than wet deposition in the open. Even if we had known this at the beginning of our experiment, the cost of isotope to simulate a 5X increase in these kinds of rates would have been prohibitive. If we could have increased dry deposition by 5X, as well as wet deposition, we might have seen more upland isotope accumulating in the fish, on an absolute basis. However, we believe it still would have been a small percentage of what we added and so would not change the general conclusions of the paper.

3. In one of their key papers on the ELA spike experiment following the spike loading phase the authors concluded 'However, a full response will be delayed by the gradual export of mercury stored in watersheds. The rate of response will vary among lakes depending on the relative surface areas of water and watershed.' And 'We expect that, in the long term, the full effect of the increased mercury loading will be much larger than the 30–40% increase in methylmercury in the biota that was evident after only 3 years.' (Harris et al., 2007). Why is this finding is no longer discussed in the present MS?

The cumulative effect of another four years of loading to Lake 658, after the first three years in the Harris et al. (2007) publication, was an increase of ~40-60% in MeHg in the biota. This is shown in Figure 1c and the overall increases in the lake ecosystems are described in the text (lines 122-132). Our wording of "the full effect" in Harris et al. (2007) was primarily

meant to refer to the fact that three years was not nearly enough time to achieve whole ecosystem steady state, especially given the slow response of the watershed.

We can now say that with 12 additional years of data since publication of Harris et al. (2007), still almost none of the watershed Hg isotopes have entered the lake and been bioaccumulated by fish. This finding is presented in Extended Data Figure 1 and stated in the main text (lines 98-107). We now reiterate and extend what was said in Harris et al. (2007) on lines 214-229. This final paragraph of the paper now speaks to the importance of the watershed in the recovery of fish from MeHg contamination.

An interesting feature of the cited long Hg residence time in watersheds is visible in N-American lake sediment cores, where Hg sedimentation rates over the past 40 years is constant (Fitzgerald et al., 2005; Engstrom et al., 2014), while Hg emission, atmospheric Hg concentrations and Hg wet deposition have dropped by a factor of 2. In other words, watershed contributions to lake Hg (and MeHg) appear to be dominant, and recovery is rather slow.

We agree with the Referee that watershed inputs of Hg to lakes are important. We now state in the text that these inputs are the dominant source of inorganic Hg to the study lake (lines 116-119; 217-218). We also agree that recovery of MeHg in fish can be slowed by the long residence time of Hg deposited to the terrestrial catchments, which we predicted would be the case for the METAALICUS watershed very early on in the study (Harris et al., 2007). We have addressed this concern specifically in our final paragraph, where we more broadly discuss the implications of the rapid response of fish MeHg to any Hg load entering the lake – whether it be from direct deposition or runoff from terrestrial areas (lines 214-229).

Also in this revised version of the paper, we emphasize the upland isotope data a bit more (Extended Data Figure 1), and point out that while the response in fish MeHg was small (lines 100-103), it clearly increased with loading and decreased when loading stopped (lines 200-201). Therefore, we can conclude that present-day Hg reductions to either the watershed or to the lake surface will have future benefits to fish and fish consumers. This is because the fish response will be immediate whenever these lower Hg loads reach the lake. This information is all presented in the final paragraph of this revised version of the paper (lines 214-229).

4. This is a spike study on one single lake; foodweb MeHg dynamics depend on the full biogeochemistry of the watershed and lake: climate, C-cycling and OM quality; soil and lake pH; biodiversity etc etc. It is common to address important questions such as ecosystem recovery or Hg re-emissions based on the review of dozens to hundreds of studies. The ELA study alone cannot fulfill that criteria unfortunately.

As I mentioned above I tremendously respect the time and energy invested in the ELA project; it was designed as best as one could at that time. I am however not convinced, based on the approach and results presented, that the swift freshwater ecosystem recovery is a robust and globally representative observation.

This realism versus replication argument has been shown time and again to be unfounded. For example, experiments done at ELA have proven to be broadly applicable to understanding eutrophication and acidification in lakes in general. We agree with the Referee that a single experiment alone cannot address all aspects of major environmental issues, such as recovery from Hg pollution, nor are they meant to. However, whole-ecosystem studies have incredible power because they manipulate only *one factor*, and in doing so provide unparalleled opportunity to understand clearly the problem under investigation. We are not saying that all lakes will respond exactly like our study lake, due to differences such as the Referee mentions. We are saying that there is a fundamental connection between mercury input to a lake and MeHg in fish. As we note in the introduction to the paper, most studies in nature are never able to make this fundamental connection because of many other confounding influences. This is the reason that dozens or more systems must be used, as the Referee notes, to try to make this type of fundamental connections from lakes where there is no single factor manipulation. Our rewrite of the final paragraph (lines 214-229) directly addresses this comment.

We interpret this main criticism – that our results are not globally representative of freshwater ecosystems – in part as a result of our not providing a broader extension of our findings from the lake to also include the entire ecosystem (i.e., the wetland and upland areas). Our focus on the lake, and specifically the response of fish MeHg to reductions in inorganic Hg loading to the lake, was due to the fact that this was the dominant source of new Hg that we could measure in biota, and most representative of how fish MeHg would respond to Hg pollution controls. The answer to that question is fish MeHg would respond almost immediately to lower inorganic Hg loads to the lake, be they from direct deposition or from runoff.

Detailed comments:

Title; suggest to formulate as ‘..recovery of mercury-contaminated fish in boreal lakes’

We don’t feel the need to specify “boreal lakes” in the title; however, we have now included this detail in the Abstract (line 56), in the paragraph describing the study site (lines 80, 92), and elsewhere in the text (lines 104, 181).

L106 ‘Delivery of lake spike inorganic Hg to sites of methylation, including sediments and anoxic bottom waters,..’

Was there evidence of methylation in the oxic lake water column? Is the fraction of foodweb MeHg originating from catchment runoff, water column methylation, and sediment methylation known? Scandinavian work on boreal lakes suggest a dominant role for catchment runoff (ex: (Bravo et al., 2017)

Please see our earlier reply to point#2 by this Referee.

L96 ‘The study catchment is located in a remote region of Canada that experiences low levels of atmospheric pollution, such that our experimental loading rate increased wet Hg deposition ~5-fold (from ~3.6 to ~19 $\mu\text{g m}^{-2} \text{y}^{-1}$), similar to more polluted regions of the world22.’

Remote lake sediment cores across N-America generally show 3x enrichment since 1850, so I'm not sure 'low level of pollution' is appropriate.

The original wording conveys the relatively lower rates of Hg deposition at the ELA compared to other areas in North America and globally. We agree that this may be confusing when considering overall levels of Hg enrichment over the past ~200 years. We have slightly modified the wording here (lines 95-98), and cited a recent study from the ELA stating: "The annual wet loadings of THg at our remote ELA site were 2.6 times lower than those in more populated areas of the nearby Great Lakes region and its subregions" (St. Louis et al. 2019)

L98 'Most of the Hg added to the upland and wetland areas surrounding Lake 658 either remained bound to vegetation and soils or evaded back to the atmosphere^{23,24}. The Hg applied to the upland catchment accounted for only a small fraction (<1%) of all Hg in runoff to the lake¹⁹ and consequently contributed little (maximum <5%) to overall fish MeHg concentrations throughout the study (Extended Data Fig. 1).'

I am not surprised by the slow spike Hg release from the catchment, but I am puzzled why ambient inorganic Hg and MeHg in catchment runoff are subsequently dismissed as an important source to lake food web MeHg. Visibly, 99% of inorganic Hg in runoff from the catchment, typically bound to DOC in boreal environment, is sustained and transiting into the lake where it is available for methylation. Adding a spike upland leads to a strong disequilibrium, and it may take decades before that spike shows up downstream; this doesn't mean that upland Hg is not an important source.

As we describe in response to an earlier comment by this Referee (point #4), catchment MeHg is not an important source of MeHg to the lake food web in our study lake. This is a factor that will be different in different watersheds, though (e.g. St. Louis et al. 1994, 1996; Bravo et al. 2017). On the other hand, upland areas are an important source of inorganic Hg to many lakes, including the METAALICUS study lake (Harris et al. 2007), which is now reiterated in this revised version (lines 116-119). It was not our intent to dismiss this potentially important source of Hg to lake food web MeHg. We now discuss the importance of catchment-derived Hg as a substrate for production of ambient MeHg to the lake food web and include a revised summary paragraph that now speaks to the importance of the watershed in the recovery of fish from MeHg contamination (lines 214-229).

Nature reviewer checklist: Raw data for Figures and discussion is not provided; Data statistics are however provided in the SI and are appropriate. Error bars are missing in most Figures or legends. Previous literature is correctly cited. MS is clear and accessible.

We plan to include the raw data for all figures in EXCEL spreadsheets for the final version. For Figure 1b,c and Figure 2 we have chosen to not include error bars. These figures are sufficiently busy without the inclusion of error bars. We provide error estimates for all fish data in Extended Data Tables 2 and 3.

References cited

- Bravo, A. G., Bouchet, S., Tolu, J., Björn, E., Mateos-Rivera, A., and Bertilsson, S.: Molecular composition of organic matter controls methylmercury formation in boreal lakes, *Nat. Commun.*, 8, 14255, <https://doi.org/10.1038/ncomms14255>, 2017.
- Engstrom, D. R., Fitzgerald, W. F., Cooke, C. A., Lamborg, C. H., Drevnick, P. E., Swain, E. B., Balogh, S. J., and Balcom, P. H.: Atmospheric Hg Emissions from Preindustrial Gold and Silver Extraction in the Americas: A Reevaluation from Lake-Sediment Archives, *Environ. Sci. Technol.*, 48, 6533–6543, <https://doi.org/10.1021/es405558e>, 2014.
- Fitzgerald, W. F., Engstrom, D. R., Lamborg, C. H., Tseng, C.-M., Balcom, P. H., and Hammerschmidt, C. R.: Modern and Historic Atmospheric Mercury Fluxes in Northern Alaska: Global Sources and Arctic Depletion, *Environ. Sci. Technol.*, 39, 557–568, <https://doi.org/10.1021/es049128x>, 2005.
- Harris, R. C., Rudd, J. W. M., Amyot, M., Babiarz, C. L., Beaty, K. G., Blanchfield, P. J., Bodaly, R. A., Branfireun, B. A., Gilmour, C. C., Graydon, J. A., Heyes, A., Hintelmann, H., Hurley, J. P., Kelly, C. A., Krabbenhoft, D. P., Lindberg, S. E., Mason, R. P., Paterson, M. J., Podemski, C. L., Robinson, A., Sandilands, K. A., Southworth, G. R., Louis, V. L. S., and Tate, M. T.: Whole-ecosystem study shows rapid fish-mercury response to changes in mercury deposition, *Proc. Natl. Acad. Sci. U. S. A.*, 104, 16586–16591, <https://doi.org/10.1073/pnas.0704186104>, 2007.
- Jonsson, S., Skjellberg, U., Nilsson, M. B., Lundberg, E., Andersson, A., and Björn, E.: Differentiated availability of geochemical mercury pools controls methylmercury levels in estuarine sediment and biota, *Nat. Commun.*, 5, 4624, <https://doi.org/10.1038/ncomms5624>, 2014.
- St. Louis, V. L., Graydon, J. A., Lehnherr, I., Amos, H. M., Sunderland, E. M., St. Pierre, K. A., Emmerton, C. A., Sandilands, K., Tate, M., Steffen, A., and Humphreys, E. R.: Atmospheric Concentrations and Wet/Dry Loadings of Mercury at the Remote Experimental Lakes Area, Northwestern Ontario, Canada, *Environ. Sci. Technol.*, 53, 8017–8026, <https://doi.org/10.1021/acs.est.9b01338>, 2019.

Response references:

- Bower, P. M., Kelly, C. A., Fee, F. J., Shearer, J. A., De-Clercq, D. R. & Schindler, D. W. Simultaneous measurement of primary production by whole-lake and bottle radiocarbon additions. *Limnol. Oceanogr.* 32, 299–312 (1987). <https://doi.org/10.4319/lo.1987.32.2.0299>
- Clarisse, O. & Hintelmann, H. Measurements of dissolved methylmercury in natural waters using diffusive gradients in thin film (DGT). *J. Environ. Monit.* 8, 1242–1247 (2006). <https://doi.org/10.1039/B614560D>

Eckley, C. S. & Hintelmann, H. Determination of mercury methylation potentials in the water column of lakes across Canada. *Sci. Total Environ.* 368, 111–125 (2006). <https://doi.org/10.1016/j.scitotenv.2005.09.042>

Haitzer, M., Aiken, G. R. & Ryan, J. N. Binding of mercury(II) to dissolved organic matter: the role of the mercury-to-DOM concentration ratio. *Environ. Sci. Technol.* 36, 3564–3570 (2002). <https://doi.org/10.1021/es025699i>

Madenjian, C. P., Chipps, S. R. & Blanchfield, P. J. Time to refine mercury mass balance models for fish. *FACETS* 6: 272–286 (2021). <https://doi.org/10.1139/facets-2020-0034>

Paterson, M. J., Blanchfield, P. J., Podemski, C. L., Hintelmann, H. H., Harris, R., Ogrinc, N., Rudd, J. W. M. & Sandilands, K.A. Bioaccumulation of newly-deposited mercury by fish and invertebrates: an enclosure study using stable mercury isotopes. *Can. J. Fish. Aquat. Sci.* 63: 2213–2224 (2006). <https://doi.org/10.1139/f06-118>

Quay, P. D., Broecker, W. S., Hesslein, R. H., Fee, E. J. & Schindler, D. W. Whole lake tritium spikes to measure horizontal and vertical mixing rates. In: *Isotopes in Lake Studies* (International Atomic Energy Agency, Vienna), pp 175–194 (1979).

Sanders, J. P., McBurney, A., Gilmour, C. C., Schwartz, G. E., Washburn, S., Kane Driscoll, S. B., Brown, S. S. & Ghosh, U. Development of a novel equilibrium passive sampling device for methylmercury in sediment and soil porewaters. *Environ. Toxicol. Chem.* 39, 323–334 (2020). <https://doi.org/10.1002/etc.4631>

Sellers, P., Kelly, C. A. & Rudd, J. W. M. Fluxes of methylmercury to the water column of a drainage lake: The relative importance of internal and external sources. *Limnol. Oceanogr.* 46, 623–631 (2001). <https://doi.org/10.4319/lo.2001.46.3.0623>

St. Louis, V. L., Rudd, J. W. M., Kelly, C. A., Beaty, K. G., Bloom, N. S. & Flett, R. J. The importance of wetlands as sources of methyl mercury to boreal forest ecosystems. *Can. J. Fish. Aquat. Sci.* 51, 1065–1076 (1994). <https://doi.org/10.1139/f94-106>

St. Louis, V. L., Rudd, J. W. M., Kelly, C. A., Beaty, K. G., Flett, R. J. & Roulet, N. T. Production and loss of methylmercury and loss of total mercury from boreal catchments containing different types of wetlands. *Environ. Sci. Technol.* 30, 2719–2729 (1996). <http://dx.doi.org/10.1021/es950856h>

Vet, R., Artz, R. S., Carou, S., Shaw, M., Ro, C-U., Aax, W., Baker, A., Bowersox, V. C., Dentener, F., Galy-Lacaux, C., Hou, A., Pienaar, J. J., Gillett, R., Forti, M. C., Gromov, S., Hara, H., Khodzher, T., Mahowald, N. M., Nickovic, S., Rao, P. S. P. & Reid, N. W. A global assessment of precipitation chemistry and deposition of sulfur, nitrogen, sea salt, base cations, organic acids, acidity and pH, and phosphorus. *Atmos. Environ.* 93, 3-100 (2014). <https://doi.org/10.1016/j.atmosenv.2013.10.060>

Reviewer Reports on the First Revision:

Referee #1 (Remarks to the Author):

Review of Blanchfield et al. "Experimental evidence for the recovery of mercury-contaminated fish"

General comments.

I thought the authors' adequately addressed the first round of comments from all three reviewers. However, when I re-read the revised manuscript, I noted a number of problems that I do not recall with the first iteration.

1) Main question: The authors state that the main question addressed is " ..whether fish MeHg respond to reductions in Hg emissions..." but I think we already know that they do and further the authors are conflating emissions and deposition here. The ecosystem experiment is simulating changes in loading not emissions so I think this is an important correction. These data are unique in that they are providing a tracer for Hg as it is methylated and bioaccumulated in the fish. I think a better phrasing of the main question would be something like: What is the a) magnitude and b) timing of fish mercury concentration changes with changes in atmospheric loading.

2) Magnitude and timing of fish mercury concentrations: I think it would greatly strengthen the paper to provide clearer quantitative information on the changes in atmospheric deposition (isotope spike) and changes in fish MeHg (spike) for the full 15 years of the experiment. This is easy for the recovery phase of the experiment since isotopic loadings have been eliminated (100% loading reduction) and the fish change by varying amounts each year.

3) Legacy mercury: the manuscript now mentions legacy mercury in several places and implications for fish Hg but this is not something is defined or discussed or that was directly observed in the work. I therefore recommend omitting it.

Specific comments:

Line 49: "toxin" typically means a biological poison vs. inorganic pollutant that should be referred to as a "toxicant"

Line 51: I agree with the other reviewer who preferred bioaccumulation to biomagnification. Bioaccumulation includes biomagnification so I am not sure why the authors objected (BAF = BMF + BCF)

Line 53: "problematic" seems like the wrong word. Suggest replacing with "challenging" or "complex"

Line 54: Here it would be helpful to indicate the ecosystem type (i.e., freshwater boreal lake)

Line 55: As mentioned above – this seems like the wrong question to me. Instead of "if fish MeHg concentrations respond solely to reductions in Hg inputs..." How about: "to determine the magnitudes and timescales of responses to reductions in Hg inputs..."

Line 58/59: Here be clearer: A 100% reduction in loading of the isotopic spike resulted in >85% reduction in prey fish and 38-78% reduction in predatory fish within 8 years. It may be helpful to highlight that this refers to the population rather than individual fish in the abstract because I was confused about this for a while.

Line 62. The study really didn't show anything about emissions controls. It shows with declines in loading, expected from emissions controls, there is a rapid reduction in fish MeHg. The study does not show anything about benefits to fish consumers either so I recommend deleting this as well. It is implied from the first few statements about the problem.

Line 64. The Convention was signed in 2013 so it is no longer that new – suggest deleting

"recently minted"

Line 65. Suggest replacing "toxic effects of MeHg" with "adverse effects of MeHg on health"

Line 66: Suggest replacing: "...which should then decrease the production of MeHg in aquatic environments" with "...which should then decrease deposition of anthropogenic Hg to aquatic environments."

Line 67-69. There is already substantial evidence for declines in fish Hg with declining loading to ecosystems. What is lacking is mechanistic and quantitative data showing the proportion and timing of such changes in response to unambiguous shifts in loading. This is what is unique about this study and these data. I think this could be better framed here. I think the part about legacy Hg should be deleted – it will confuse readers and is not defined. The important point is there is a lot of Hg in the environment already so sometimes the magnitude and timing of changes in response to shifts in loading is not clear. Then the further complicating factors can be discussed.

Line 73. Climate changes could also dampen MeHg production under certain conditions so I suggest "alter" instead of "enhance."

Line 78: Suggest listing the full duration of the experiment here: 2001-2016 (?)

Line 87: Replace "emissions" with "deposition"

Line 96: Why "even the Great Lakes" – the Great Lakes are typically thought of as industrially contaminated areas – at least some of them – so I suggest rephrasing this.

Line 103: Suggest deleting "future" from this sentence – implies it will happen in this lake.

Line 110: If "spike MeHg" is used to refer to the MeHg measured that was produced from the labeled Hg it should be used consistently here.

Lines 116-121. I am confused by the logic here. The authors state ambient watershed Hg inputs are relatively stable and this is why the ambient fish Hg concentrations are stable. Elsewhere and a couple lines below they say what accumulates in the fish is the atmospherically deposited Hg rather than the watershed Hg, presumably reflecting both a lack of large inputs and differences in bioavailability. Isn't it therefore possible that the ambient fish Hg trends are also dictated by ambient atmospheric deposition directly to the lake not the watershed loading, even if the watershed loading is a substantial fraction of the overall mass of Hg inputs? The second part of the paragraph says evidence for ambient in-lake methylation is provided by methylation of the spike seasonally but this seems a little circular to me. I think the more logical order of ideas would be: 1) The only source of spike MeHg was in-lake methylation, 2) Seasonal patterns in spike MeHg and ambient MeHg varied synchronously in the lake, and 3) this suggests ambient MeHg is also mainly from in-lake methylation.

Lines 122-132: Here and elsewhere where these types of %changes are provided, a discussion comparing the changes in atmospheric loading to those of water, sediment, and fish would be very helpful. Otherwise, the changes have no reference point and study results are very qualitative.

Lines 130-132: It is expected that predators lag changes in prey. The way it phrased makes it sound like a novel finding.

Line 194-195. Suggest deleting the part about legacy Hg and rephrasing this. I don't agree with this statement and legacy Hg was not the focus of this work so any conclusions related to legacy Hg are not directly observed. This experiment convincingly shows that despite a lot of ambient Hg (natural plus previously deposited anthropogenic) present in the ecosystem, fish concentrations

change rapidly in response to new loading reductions.

Lines 211-213: This sentence should be deleted in my opinion because text suggests this experiment can be used as the basis for inferring the types of changes in other ecosystems but this will be a function of the properties of those ecosystems.

Line 220-226. This is hard to follow and should be rephrased for clarity. "inertia of the watershed" is particularly ambiguous.

Line 226-227. Note that these models do not include any processes related to methylation and fish bioaccumulation and simply scale everything linearly with atmospheric deposition.

Line 227-229: This study does not provide any information on cost effectiveness – this should be deleted.

Figure 1: I am confused about what the fish represent here. Please add information on sample size and whether age cohorts were tracked individually or this is a "population" average. This is important for understanding Figure 3.

Figure 2: Suggest making the spike stand out more with color. The legend is very small and in the corner so this could be moved to the top of the figure as well so it is not missed. Is the BMF information needed as part of the main text? This seems like a secondary point to me and makes it harder to see the loading/recovery changes in concentrations.

Figure 3. I think I finally understand that there was longitudinal sampling of the pike – which is very cool – but somewhat disconnected from the narrative of the paper and the figure is showing two contrasting things – which is very confusing. As I understand it, the population of pike (including new individuals) is declining in lake spike burden (simple point and consistent with the rest of the paper) but the individual fish are not significantly changing over time. It would be nice to see the individual fish in a separate panel and with connected lines to show temporal changes in individuals more clearly. This also warrants more discussion in the main text. Large predatory fish themselves appear not to eliminate the spike (are they at an age where there is no longer growth dilution?). This seems to merit a bit more discussion. This actually seems like very important data to me because you can say something about the temporal dynamics of individual fish (at least pike) – so please expand this. There are other things that can be condensed in the existing text in my opinion that would make room for these exciting data and a little discussion.

Referee #2 (Remarks to the Author):

My comments are very minor and I think the authors did a very good job of addressing comments. My only somewhat major comment is around line 60 to 63. I wish the authors would not generalize this much and, and I did not like the word "alone" on line 62. It is a very strong conclusion that I feel is only really supported by the data in this study for lakes where in-lake methylation is the dominant form of MeHg getting into the food web and the major Hg load increase is from the atmosphere. The reason I bring this up is that this is the sort of sentence in a Nature article that could be misunderstood and used to argue against places where Hg loadings to lakes might increase because of land use change or melting permafrost, but the regulators are really only hung up on atmospheric Hg loadings. I liked how the decline was better explained in the final paragraph "The spike MeHg show that fish MeHg concentrations will respond quickly to any change in loading rates – be they from direct deposition to the lake or runoff – so as these two loads decrease, the fish populations in the receiving lake will soon afterwards have lower MeHg than they would have if nothing were done."

Minor comments:

Line 55: I don't think the word solely is necessary. I get they are trying to address the reviewer comments, but this will be elaborated upon in the paper and I find it an odd inclusion here.

Line 86: I think the reduction was just one of the key questions? I would guess... I think there were many questions, so maybe qualify this statement.

Line 146: "will" maybe should be "would"

Referee #3 (Remarks to the Author):

In their revised version Blanchfield et al. have addressed in detail the comments and questions raised by all reviewers. I appreciate the discussion on the lake spike strategy and spike bioavailability in light of my questions and agree that the synchrony between ambient and spiked Hg indicates similar in-lake reactivity and methylation. Their responses have also clarified a number of details on the dominant inorganic Hg (watershed, but not atmosphere) and MeHg (formed in-lake, little from watershed) sources to the lake ecosystem. In particular the inorganic Hg source to the lake (watershed or atmosphere), and to lakes in general, is a critical control factor on the potential rapidity of ecosystem recovery following cuts in Hg emissions. On this particular key issue, the rapid timing of recovery, I disagree with the authors' arguments and rebuttal; I will illustrate this via various phrases in the MS:

The following phrase (L117) explicitly clarifies that watershed inputs are the main Hg source to lake 658 (75%, see Harris et al., 2007 Fig 3):

L117 "Steady ambient MeHg concentrations in fish through time are indicative of relatively stable watershed inputs of Hg, which is the main source of ambient inorganic Hg to both the experimental and reference lakes19,26."

The revised closing paragraph (L221-229 below) revisits the importance of the watershed dominated inorganic Hg source to the lake, and correctly argues that any change in loading rate from watershed and from atmosphere will lead to rapid fish MeHg changes. The paragraph acknowledges the importance of the inertia of the watershed: this means that for lakes with dominant watershed inputs (i.e. ELA lakes and most lakes in the world), a decrease in Hg emissions will not immediately cause a decrease in Hg loading to lakes, because of a lagged response that continues to deliver legacy Hg, deposited since pre-industrial times in the watershed, to lakes. The authors' wording 'as these two loads decrease' suggests synchrony in the minor atmospheric Hg load and the dominant watershed Hg load upon decrease global Hg emissions. This is questionable and in contradiction with the known inertia of the watershed Hg pool. As in my 1st review I cite the author's own words in Harris et al. 2007: "However, a full response will be delayed by the gradual export of mercury stored in watersheds. The rate of response will vary among lakes depending on the relative surface areas of water and watershed." The authors then cite 'current models' (L226) that predict rapid (decades) benefits, yet these models directly parameterize ecosystem loading and human exposure as a function of atmospheric inputs only and therefore ignore the inertia of terrestrial watersheds in delivering inorganic Hg to lakes, wetlands and coastal waters. It is therefore correct that the authors' findings support those models, with the caveat that both models and this study ignore watershed inertia and are therefore not addressing the main control factor on ecosystem recovery: the watershed to lake area ratio.

L221-229 "Taken together, this means that the inertia of the watershed, because of the large amounts of Hg stored there, must be overcome before Hg in watershed runoff declines in response to decreasing atmospheric deposition rates. The spike MeHg data show that fish MeHg concentrations will respond quickly to any change in loading rates – be they from direct deposition

to the lake or runoff – so as these two loads decrease, the fish populations in the receiving lake will soon afterwards have lower MeHg than they would have if nothing were done. Current models using partial Hg reduction scenarios predict major economic benefits over the coming decades from improved human health^{32,33}. Our findings provide strong support that legislation to reduce global Hg emissions will be cost effective by allowing for the recovery of contaminated fisheries, which will in turn reduce human exposure.”

If I now jump to the closing phrase of the abstract (L61 below), I find that the authors overstate that ‘immediate’ benefits will result from Hg emission control. That would be correct for a waterbody that is predominantly supplied by atmospheric Hg, based on the authors findings. However, it is incorrect for a waterbody, such as the lake 658 studied, that is mostly supplied with watershed inorganic Hg (75% watershed inputs; (Harris et al., 2007) Fig 3).

L61 “This initiated surprisingly rapid MeHg concentration declines of 38-76% in lake whitefish and northern pike populations in only eight years, clearly showing that Hg emission controls alone will have immediate and future benefits to fish consumers.”

Let me formulate my criticism as a question: if by 2030 atmospheric Hg emissions and atmospheric Hg deposition to lakes drop by a factor of 2, will the ELA fish and lake fish MeHg content in general also drop by a factor of 2? The answer, I think, is ‘we don’t know’ because the lake spike loading experiment and results presented by the authors cannot answer that question for a lake with dominant watershed Hg inputs. The watershed spike results will some day shed light on this, but the inertia may be long enough that this may take decades if not centuries.

A possible illustration of the importance of watershed processes on lake 658 is perhaps reflected in ambient MeHg in lake and fish which increased by 30-90% over the study period (Figure 2abc). Over the same study period, N-American wet deposition decreased by -1.6% per year, which amounts to -22% over the 2001-2015 study period (Zhang et al., 2016). Zhang et al. attribute the decline in Hg wet deposition (and atmospheric Hg₀) to a decline in global Hg emissions, in particular in N-America and Europe. How to reconcile the observation that decreased Hg emission and deposition led to increased fish MeHg levels in lake 658?

Minor comments:

The title and phrasing in abstract (L63) and concluding paragraph (L229) uses fish and fishery in a generic way (including freshwater and marine fisheries), whereas it should be ‘freshwater fish and fisheries’ only, because that is what is under study.

Cited literature:

Harris, R. C., Rudd, J. W. M., Amyot, M., Babiarz, C. L., Beaty, K. G., Blanchfield, P. J., Bodaly, R. A., Branfireun, B. A., Gilmour, C. C., Graydon, J. A., Heyes, A., Hintelmann, H., Hurley, J. P., Kelly, C. A., Krabbenhoft, D. P., Lindberg, S. E., Mason, R. P., Paterson, M. J., Podemski, C. L., Robinson, A., Sandilands, K. A., Southworth, G. R., Louis, V. L. S., and Tate, M. T.: Whole-ecosystem study shows rapid fish-mercury response to changes in mercury deposition, 104, 16586–16591, <https://doi.org/10.1073/pnas.0704186104>, 2007.

Zhang, Y., Jacob, D. J., Horowitz, H. M., Chen, L., Amos, H. M., Krabbenhoft, D. P., Slemr, F., St. Louis, V. L., and Sunderland, E. M.: Observed decrease in atmospheric mercury explained by global decline in anthropogenic emissions, 113, 526–531, <https://doi.org/10.1073/pnas.1516312113>, 2016.

Author Rebuttals to First Revision:

Referee #1 (Remarks to the Author):

Review of Blanchfield et al. “Experimental evidence for the recovery of mercury-contaminated fish”

General comments.

I thought the authors’ adequately addressed the first round of comments from all three reviewers. However, when I re-read the revised manuscript, I noted a number of problems that I do not recall with the first iteration.

We thank the Referee for their initial round of comments. Below we respond to their new review.

1) Main question: The authors state that the main question addressed is “..whether fish MeHg respond to reductions in Hg emissions...” but I think we already know that they do and further the authors are conflating emissions and deposition here. The ecosystem experiment is simulating changes in loading not emissions so I think this is an important correction. These data are unique in that they are providing a tracer for Hg as it is methylated and bioaccumulated in the fish. I think a better phrasing of the main question would be something like: What is the a) magnitude and b) timing of fish mercury concentration changes with changes in atmospheric loading.

We agree that it is most correct to call this a loading experiment, and we have changed “emissions” to “loading” when describing the main goal and findings of this study (lines 56, 90, 202). Elsewhere, we have distinguished “emissions” and “loading” more obviously from each other for understanding the influence of emission controls on Hg loading to lakes and MeHg in fish (lines 65, 199).

We disagree that “*we already know*” that fish MeHg will decrease in any reasonable time frame with a decrease in Hg loading. For example, a recent study of multiple data sets concluded that this relationship is sometimes seen and sometimes not (Wang et al. 2019). The unique design of the METAALICUS experiment is that it changed only one variable – Hg loading – and clearly showed that the MeHg concentrations in the fish populations responded quickly to the reduced loading. We do agree with the suggestion by the Referee about rephrasing the main question of the study and have revised the text to include the magnitude and timing of the response of fish MeHg concentrations to reductions in Hg loading in the Abstract (lines 54-56) and in the main body of the article (lines 89-92, 157-160).

2) Magnitude and timing of fish mercury concentrations: I think it would greatly strengthen the paper to provide clearer quantitative information on the changes in atmospheric deposition (isotope spike) and changes in fish MeHg (spike) for the full 15 years of the experiment. This is easy for the recovery phase of the experiment since isotopic loadings have been eliminated (100% loading reduction) and the fish change by varying amounts each year.

We have included a sentence stating that we roughly doubled the Hg load to the lake (i.e. increase by 100%) from the annual additions of lake spike Hg (as reported in Harris et al. 2007). This gives a reference point for the % increases and decreases in fish that resulted (lines 128-129).

3) Legacy mercury: the manuscript now mentions legacy mercury in several places and implications for fish Hg but this is not something is defined or discussed or that was directly observed in the work. I therefore recommend omitting it.

Given the suggestions by Referee #3 to provide information about Hg inputs to the lake from the watershed, which is mainly older Hg, we have chosen to retain the text related to “legacy Hg”. However, because the term “legacy Hg” is sometimes used to refer to past point source inputs of Hg, which is not appropriate in the context of this study, in each place where this word was used, we now use more specific explanatory terms (lines 69, 84, 189).

Specific comments:

Line 49: “toxin” typically means a biological poison vs. inorganic pollutant that should be referred to as a “toxicant”

We made this change.

Line 51: I agree with the other reviewer who preferred bioaccumulation to biomagnification. Bioaccumulation includes biomagnification so I am not sure why the authors objected (BAF = BMF + BCF)

We made this change.

Line 53: “problematic” seems like the wrong word. Suggest replacing with “challenging” or “complex”

We now use the word “complex”.

Line 54: Here it would be helpful to indicate the ecosystem type (i.e., freshwater boreal lake)

We have now included “freshwater” to define the type of fish (line 55) with “boreal lake” stated in the same sentence (line 56).

Line 55: As mentioned above – this seems like the wrong question to me. Instead of “if fish MeHg concentrations respond solely to reductions in Hg inputs...” How about: “to determine the magnitudes and timescales of responses to reductions in Hg inputs...”

We have changed the wording as suggested (see earlier response to comment #1).

Line 58/59: Here be clearer: A 100% reduction in loading of the isotopic spike resulted in >85% reduction in prey fish and 38-78% reduction in predatory fish within 8 years. It may be helpful to highlight that this refers to the population rather than individual fish in the abstract because I was confused about this for a while.

We have revised the Abstract to make it clear that cessation of loading emulated a 100% Hg reduction scenario and that when we refer to the % declines in the labeled MeHg concentrations in fish, we are referring to population-level data only (lines 58-60).

Line 62. The study really didn't show anything about emissions controls. It shows with declines in loading, expected from emissions controls, there is a rapid reduction in fish MeHg. The study does not show anything about benefits to fish consumers either so I recommend deleting this as well. It is implied from the first few statements about the problem.

These changes have been made as recommended.

Line 64. The Convention was signed in 2013 so it is no longer that new – suggest deleting “recently minted”

We made this change.

Line 65. Suggest replacing “toxic effects of MeHg” with “adverse effects of MeHg on health”

We replaced with “adverse effects of MeHg”.

Line 66: Suggest replacing: “...which should then decrease the production of MeHg in aquatic environments” with “...which should then decrease deposition of anthropogenic Hg to aquatic environments.”

We changed the wording to: “...which should then decrease deposition and loading of anthropogenic Hg to aquatic environments.”

Line 67-69. There is already substantial evidence for declines in fish Hg with declining loading to ecosystems. What is lacking is mechanistic and quantitative data showing the proportion and timing of such changes in response to unambiguous shifts in loading. This is what is unique about this study and these data. I think this could be better framed here. I think the part about legacy Hg should be deleted – it will confuse readers and is not defined. The important point is there is a lot of Hg in the environment already so sometimes the magnitude and timing of changes in response to shifts in loading is not clear. Then the further complicating factors can be discussed.

Please refer to our earlier response to comments #1, #2 and #3. We have deleted “legacy” where used, and substituted other more specifically defined terminology.

Line 73. Climate changes could also dampen MeHg production under certain conditions so I suggest “alter” instead of “enhance.”

We replaced “enhance” with “influence” (line 74).

Line 78: Suggest listing the full duration of the experiment here: 2001-2016 (?)

We included this information (line 79).

Line 87: Replace “emissions” with “deposition”

We have replaced “emissions” with “loading” here (line 90). Please see also earlier response to comment #1.

Line 96: Why “even the Great Lakes” – the Great Lakes are typically thought of as industrially contaminated areas – at least some of them – so I suggest rephrasing this.

We deleted the reference to the Great Lakes from this sentence (lines 97-99).

Line 103: Suggest deleting “future” from this sentence – implies it will happen in this lake.

We made this change (line 105).

Line 110: If “spike MeHg” is used to refer to the MeHg measured that was produced from the labeled Hg it should be used consistently here.

We made this change (line 116).

Lines 116-121. I am confused by the logic here. The authors state ambient watershed Hg inputs are relatively stable and this is why the ambient fish Hg concentrations are stable. Elsewhere and a couple lines below they say what accumulates in the fish is the atmospherically deposited Hg rather than the watershed Hg, presumably reflecting both a lack of large inputs and differences in bioavailability. Isn't it therefore possible that the ambient fish Hg trends are also dictated by ambient atmospheric deposition directly to the lake not the watershed loading, even if the watershed loading is a substantial fraction of the overall mass of Hg inputs?

We thank the Referee for pointing out a potential source of confusion. We have rewritten the last sentence to make it clear that the largest inputs of Hg to the lake are from the watershed (lines 122-125), and that this watershed Hg is methylated in the lake and is the dominant source of *ambient* MeHg in fish (lines 109-113).

We can't find anywhere in the manuscript where we stated that “what accumulates in the fish is the atmospherically deposited Hg rather than the watershed Hg” and can only think that the Referee is mixing up the spike results with the *ambient* (i.e., the direct deposit lake spike Hg was the main source of spike MeHg in fish, not the watershed spike Hg). Again, we think that the revised simpler sentence at the end of the paragraph is clearer on this (lines 122-125).

In the lake and biota, seasonal variation in MeHg concentrations of the lake spike showed high temporal synchrony (Figure 4 from Harris et al. 2007), which we interpret as a similar bioavailability of atmospherically-deposited Hg (lake spike in our study) and watershed Hg (ambient). See next comment.

We agree that our findings support the contention that ambient fish MeHg trends are also influenced by the amount of ambient Hg falling directly onto the lake surface, irrespective of watershed inputs (see Figure 2 a-c). The influence of direct Hg loadings onto the lake on

overall fish MeHg concentrations will be most pronounced in lakes receiving most of their Hg load from the atmosphere compared to lakes receiving most of their Hg from their watershed. We now include a statement about surface area:lake watershed area (lines 197-200).

The second part of the paragraph says evidence for ambient in-lake methylation is provided by methylation of the spike seasonally but this seems a little circular to me. I think the more logical order of ideas would be: 1) The only source of spike MeHg was in-lake methylation, 2) Seasonal patterns in spike MeHg and ambient MeHg varied synchronously in the lake, and 3) this suggests ambient MeHg is also mainly from in-lake methylation.

We agree with this reordering of ideas. We also think that this sentence was confusing in its placement at the end of this paragraph and have therefore moved this piece of evidence to the end of the previous paragraph, where we think it makes more sense (lines 109-113).

Lines 122-132: Here and elsewhere where these types of %changes are provided, a discussion comparing the changes in atmospheric loading to those of water, sediment, and fish would be very helpful. Otherwise, the changes have no reference point and study results are very qualitative.

The following text has been added (lines 128-130): “The addition of lake spike Hg was roughly equivalent to all ambient Hg inputs (runoff plus direct deposition) to the lake, resulting in a doubling, or ~100% increase, in Hg loading¹⁹. In response, ...”. This is followed by the % increases in MeHg in the different compartments (text unchanged). See also our earlier response to comment #2.

Lines 130-132: It is expected that predators lag changes in prey. The way it phrased makes it sound like a novel finding.

We have removed this sentence from the manuscript. The following paragraph on biomagnification (lines 136-141) adequately captures the lagged response of the different fish species to their prey.

Line 194-195. Suggest deleting the part about legacy Hg and rephrasing this. I don't agree with this statement and legacy Hg was not the focus of this work so any conclusions related to legacy Hg are not directly observed. This experiment convincingly shows that despite a lot of ambient Hg (natural plus previously deposited anthropogenic) present in the ecosystem, fish concentrations change rapidly in response to new loading reductions.

The term “legacy” has been deleted and replaced with “previously deposited” (please see earlier response to comment #3) (line 189).

Lines 211-213: This sentence should be deleted in my opinion because text suggests this experiment can be used as the basis for inferring the types of changes in other ecosystems but this will be a function of the properties of those ecosystems.

We agree with this comment. We have removed the text (formerly lines 207-213) dealing with the rate of change in ecosystem response to loading versus recovery (including the

removal of Extended Data Figure 3). Some reductions in word count to the manuscript were necessary to allow for the additional information requested by the Referees, and we felt this information was secondary to the main theme of the response of fish MeHg concentrations to reductions in Hg loading.

Line 220-226. This is hard to follow and should be rephrased for clarity. “inertia of the watershed” is particularly ambiguous.

The final paragraph of the paper has been rewritten in response to more than one of the Referee’s comments. Namely to address the differentiation between emissions, deposition, and loading. The term “inertia” was removed during the rewriting process (lines 201-208).

Line 226-227. Note that these models do not include any processes related to methylation and fish bioaccumulation and simply scale everything linearly with atmospheric deposition.

We thank the Referee for pointing out the limitations of the models in these studies. We have revised this final paragraph, and no longer cite these studies.

Line 227-229: This study does not provide any information on cost effectiveness – this should be deleted.

We have deleted reference to “cost effectiveness”.

Figure 1: I am confused about what the fish represent here. Please add information on sample size and whether age cohorts were tracked individually or this is a “population” average. This is important for understanding Figure 3.

The fish data in Figure 1 are an annual population average for each species (or age class for yellow perch). Exact sample sizes (and associated metrics) are provided in Extended Data Tables 2 and 3. We have added the requested information to the figure caption.

Figure 2: Suggest making the spike stand out more with color. The legend is very small and in the corner so this could be moved to the top of the figure as well so it is not missed. Is the BMF information needed as part of the main text? This seems like a secondary point to me and makes it harder to see the loading/recovery changes in concentrations.

We used black for lake spike so that it would stand out. The BMF data are important for: (1) demonstrating similar trophic transfer of ambient and lake spike MeHg during the loading phase; and (2) elucidating the speed of equilibrium between predator and prey during loading and recovery. We have increased the size of the legend in panel “c”.

Figure 3. I think I finally understand that there was longitudinal sampling of the pike – which is very cool – but somewhat disconnected from the narrative of the paper and the figure is showing two contrasting things – which is very confusing. As I understand it, the population of pike (including new individuals) is declining in lake spike burden (simple point and consistent with the rest of the paper) but the individual fish are not significantly changing over time. It would be nice to see the individual fish in a separate panel and with connected lines to show temporal changes in individuals more clearly. This also warrants more discussion in the main text. Large predatory fish themselves appear not to eliminate the spike (are they at an age where there is no longer growth dilution?). This seems to merit a bit more

discussion. This actually seems like very important data to me because you can say something about the temporal dynamics of individual fish (at least pike) – so please expand this.

The Referee is correct regarding the data presented in Figure 3. We feel that the inclusion of individual information is warranted because it serves to highlight two important findings critical to understand the recovery of fish populations from MeHg contamination: (1) once bioaccumulated, MeHg will remain within the muscle tissue of large-bodied fish for a very long time; and (2) the rapid decline in northern pike MeHg burdens (and concentrations) observed for *the population* is driven by population turnover – the recruitment of new fish into the population and loss of older ones.

We agree that these data could be presented more cleanly, and have taken the Referee’s advice to show the temporal changes in individual northern pike through time by connecting lines for individual fish. We have opted to not split this figure into separate panels, though, given the additional space requirements (and associated reductions in word allowance). We have also revised the Figure 3 caption for clarity.

Expressing the spike as body burden (y-axis of Figure 3) accounts for changes in lake spike MeHg concentration due to growth dilution. The annual body size data (presented in Extended Data Figure 2) show that these individual pike grew steadily throughout the recovery period.

We know of no other study that simultaneously tracked individual *and* population-level fish MeHg burdens over time, and agree that these data warranted the additional explanation now found in the revised paragraph (lines 175-188) and in the Abstract (lines 60-63).

Referee #2 (Remarks to the Author):

My comments are very minor and I think the authors did a very good job of addressing comments. My only somewhat major comment is around line 60 to 63. I wish the authors would not generalize this much and, and I did not like the word “alone” on line 62. It is a very strong conclusion that I feel is only really supported by the data in this study for lakes where in-lake methylation is the dominant form of MeHg getting into the food web and the major Hg load increase is from the atmosphere. The reason I bring this up is that this is the sort of sentence in a Nature article that could be misunderstood and used to argue against places where Hg loadings to lakes might increase because of land use change or melting permafrost, but the regulators are really only hung up on atmospheric Hg loadings. I liked how the decline was better explained in the final paragraph “The spike MeHg show that fish MeHg concentrations will respond quickly to any change in loading rates – be they from direct deposition to the lake or runoff – so as these two loads decrease, the fish populations in the receiving lake will soon afterwards have lower MeHg than they would have if nothing were done.”

We thank the Referee for their positive comments. Our inclusion of the word “alone” (and “solely”; see next comment) was to inform the reader that many of the past challenges

associated with defining a direct causal relationship between Hg loadings and MeHg concentrations in fish (described in the Introduction) were overcome in this study by using enriched stable isotopes of Hg to experimentally increase Hg loadings. We appreciate that this phrasing could be interpreted in a way that could exclude other sources of Hg inputs to lakes. We have altered the text to remove “alone”. We have also revised the final sentence of the Abstract (lines 60-63) to be more specific to the findings of our study.

Minor comments:

Line 55: I don't think the word solely is necessary. I get they are trying to address the reviewer comments, but this will be elaborated upon in the paper and I find it an odd inclusion here.

We have revised this sentence and omitted “solely”.

Line 86: I think the reduction was just one of the key questions? I would guess... I think there were many questions, so maybe qualify this statement.

The Referee is correct in that there were many key questions, but this paper focuses on the most essential one – *recovery*. We have rephrased this sentence to include the magnitude and timing of recovery in fisheries MeHg concentrations in response to reductions in Hg loading (lines 89-92; see also the response to comment #1 by Referee #1).

Line 146: “will” maybe should be “would”

We made this change (line 148).

Referee #3 (Remarks to the Author):

In their revised version Blanchfield et al. have addressed in detail the comments and questions raised by all reviewers. I appreciate the discussion on the lake spike strategy and spike bioavailability in light of my questions and agree that the synchrony between ambient and spiked Hg indicates similar in-lake reactivity and methylation. Their responses have also clarified a number of details on the dominant inorganic Hg (watershed, but not atmosphere) and MeHg (formed in-lake, little from watershed) sources to the lake ecosystem. In particular the inorganic Hg source to the lake (watershed or atmosphere), and to lakes in general, is a critical control factor on the potential rapidity of ecosystem recovery following cuts in Hg emissions. On this particular key issue, the rapid timing of recovery, I disagree with the authors' arguments and rebuttal; I will illustrate this via various phrases in the MS:

We thank the Referee for their initial round of comments that prompted us to add further details of the study. Below we address the specific issues related to ecosystem recovery.

The following phrase (L117) explicitly clarifies that watershed inputs are the main Hg source to lake 658 (75%, see Harris et al., 2007 Fig 3):

L117 “Steady ambient MeHg concentrations in fish through time are indicative of relatively stable watershed inputs of Hg, which is the main source of ambient inorganic Hg to both the experimental and reference lakes19,26.”

The revised closing paragraph (L221-229 below) revisits the importance of the watershed dominated inorganic Hg source to the lake, and correctly argues that any change in loading rate from watershed and from atmosphere will lead to rapid fish MeHg changes. The paragraph acknowledges the importance of the inertia of the watershed: this means that for lakes with dominant watershed inputs (i.e. ELA lakes and most lakes in the world), a decrease in Hg emissions will not immediately cause a decrease in Hg loading to lakes, because of a lagged response that continues to deliver legacy Hg, deposited since pre-industrial times in the watershed, to lakes.

This is an accurate recounting of the text of the manuscript. We agree that a decrease in emissions will not immediately cause a decrease in Hg loading to lakes from the watershed, although the decrease will be immediate for Hg falling directly onto the lake.

The authors’ wording ‘as these two loads decrease’ suggests synchrony in the minor atmospheric Hg load and the dominant watershed Hg load upon decrease global Hg emissions. This is questionable and in contradiction with the known inertia of the watershed Hg pool. As in my 1st review I cite the author’s own words in Harris et al. 2007: “However, a full response will be delayed by the gradual export of mercury stored in watersheds. The rate of response will vary among lakes depending on the relative surface areas of water and watershed.”

We certainly did not mean to suggest synchrony in how these two loading sources would decline over time. As stated in the Abstract of our earlier paper (Harris et al. 2007), and clearly reiterated here, we expect a bi-phasic response in fisheries MeHg concentrations to Hg pollution controls because of: 1) an initial rapid response from reductions in the deposition of Hg directly to the lake surface; and 2) a delayed response for Hg entering the lake from the watershed following deposition there. Further, the Referee characterizes the atmospheric load to our study lake as “minor”, but in fact it constituted ~20% of the annual Hg loading to the lake.

The authors then cite ‘current models’ (L226) that predict rapid (decades) benefits, yet these models directly parameterize ecosystem loading and human exposure as a function of atmospheric inputs only and therefore ignore the inertia of terrestrial watersheds in delivering inorganic Hg to lakes, wetlands and coastal waters. It is therefore correct that the authors’ findings support those models, with the caveat that both models and this study ignore watershed inertia and are therefore not addressing the main control factor on ecosystem recovery: the watershed to lake area ratio.

L221-229 “Taken together, this means that the inertia of the watershed, because of the large amounts of Hg stored there, must be overcome before Hg in watershed runoff declines in

response to decreasing atmospheric deposition rates. The spike MeHg data show that fish MeHg concentrations will respond quickly to any change in loading rates – be they from direct deposition to the lake or runoff – so as these two loads decrease, the fish populations in the receiving lake will soon afterwards have lower MeHg than they would have if nothing were done. Current models using partial Hg reduction scenarios predict major economic benefits over the coming decades from improved human health^{32,33}. Our findings provide strong support that legislation to reduce global Hg emissions will be cost effective by allowing for the recovery of contaminated fisheries, which will in turn reduce human exposure.”

The Referee is correct in that the models in the studies we cited determine human health benefits from Hg deposition reductions based on exposure from changes in atmospheric deposition only (see response to comment by Referee #1 regarding lines 226-227). We have revised the final paragraph of the paper and no longer cite these studies.

If I now jump to the closing phrase of the abstract (L61 below), I find that the authors overstate that ‘immediate’ benefits will result from Hg emission control. That would be correct for a waterbody that is predominantly supplied by atmospheric Hg, based on the authors findings. However, it is incorrect for a waterbody, such as the lake 658 studied, that is mostly supplied with watershed inorganic Hg (75% watershed inputs; (Harris et al., 2007) Fig 3).

L61 “This initiated surprisingly rapid MeHg concentration declines of 38-76% in lake whitefish and northern pike populations in only eight years, clearly showing that Hg emission controls alone will have immediate and future benefits to fish consumers.”

Let me formulate my criticism as a question: if by 2030 atmospheric Hg emissions and atmospheric Hg deposition to lakes drop by a factor of 2, will the ELA fish and lake fish MeHg content in general also drop by a factor of 2? The answer, I think, is ‘we don’t know’ because the lake spike loading experiment and results presented by the authors cannot answer that question for a lake with dominant watershed Hg inputs. The watershed spike results will some day shed light on this, but the inertia may be long enough that this may take decades if not centuries.

The manuscript has been revised to clearly say that this is a loading study and that fish will respond quickly to changes in loading. Further, it is now stated that it will take much longer than 10 years for a full response in fish MeHg content after Hg deposition is lowered because of the slow response of the watershed (lines 197-200).

In our study, we show that fish populations all contained some lake spike MeHg eight years after complete cessation of Hg loading to the lake itself (Figs. 1-3). Thus, the scenario of a 1:1 response in ELA fish MeHg content to incremental reductions in Hg deposition, over the time period when those reductions are occurring, does not allow sufficient time for fish to fully respond to the decreasing Hg load to the lake, irrespective of the watershed.

A possible illustration of the importance of watershed processes on lake 658 is perhaps reflected in ambient MeHg in lake and fish which increased by 30-90% over the study period (Figure 2abc). Over the same study period, N-American wet deposition decreased by -1.6% per year, which amounts to -22% over the 2001-2015 study period (Zhang et al., 2016). Zhang et al. attribute the decline in Hg wet deposition (and atmospheric Hg₀) to a decline in global Hg emissions, in particular in N-America and Europe. How to reconcile the observation that decreased Hg emission and deposition led to increased fish MeHg levels in lake 658?

Although wet deposition of Hg across North America has generally decreased 1.6% per year, at the ELA, wet deposition of Hg *increased* steadily between 2002-2010 by 0.35 ug/m²/y (St. Louis et al. 2019). As a point of clarification, only the lake whitefish population showed a significant increase in ambient MeHg concentration over time (2000-2015) in Lake 658 (Fig. 2c). The yellow perch (age 1+; Fig. 2a) and northern pike (Fig. 2b) populations did not (see Extended Data Table 1). The Lake 658 lake whitefish population was comprised of old, large fish. Unlike northern pike, we observed little new recruitment of young lake whitefish into the population during our annual sampling. It is most likely that the increasing ambient MeHg concentrations reflect the continued uptake of ambient MeHg with increasing age of the Lake 658 lake whitefish population over time.

Minor comments:

The title and phrasing in abstract (L63) and concluding paragraph (L229) uses fish and fishery in a generic way (including freshwater and marine fisheries), whereas it should be 'freshwater fish and fisheries' only, because that is what is under study.

We have clarified in the Abstract (line 55), in the main text (line 94), and in the concluding paragraph (line 208) that our study pertains to freshwater fish and fisheries. We have not included freshwater in the title, as this would increase its length beyond the allowed number of characters.

Cited literature:

Harris, R. C., Rudd, J. W. M., Amyot, M., Babiarz, C. L., Beaty, K. G., Blanchfield, P. J., Bodaly, R. A., Branfireun, B. A., Gilmour, C. C., Graydon, J. A., Heyes, A., Hintelmann, H., Hurley, J. P., Kelly, C. A., Krabbenhoft, D. P., Lindberg, S. E., Mason, R. P., Paterson, M. J., Podemski, C. L., Robinson, A., Sandilands, K. A., Southworth, G. R., Louis, V. L. S., and Tate, M. T.: Whole-ecosystem study shows rapid fish-mercury response to changes in mercury deposition, 104, 16586–16591, <https://doi.org/10.1073/pnas.0704186104>, 2007.

Zhang, Y., Jacob, D. J., Horowitz, H. M., Chen, L., Amos, H. M., Krabbenhoft, D. P., Slemr,

F., St. Louis, V. L., and Sunderland, E. M.: Observed decrease in atmospheric mercury explained by global decline in anthropogenic emissions, 113, 526–531, <https://doi.org/10.1073/pnas.1516312113>, 2016.

Response references:

St. Louis, V. L., Graydon, J. A., Lehnerr, I., Amos, H. M., Sunderland, E. M., St. Pierre, K. A., Emmerton, C. A., Sandilands, K., Tate, M., Steffen, A. & Humphreys, E. R. Atmospheric concentrations and wet/dry loadings of mercury at the remote Experimental Lakes Area, Northwestern Ontario, Canada. *Environ. Sci. Technol.* 53, 8017–8026 (2019). DOI: 10.1021/acs.est.9b01338

Wang, F., Outridge, P. M., Feng, X., Meng, B., Heimbürger-Boavida, L-E. & Mason, R. P. How closely do mercury trends in fish and other aquatic wildlife track those in the atmosphere? – Implications for evaluating the effectiveness of the Minamata Convention. *Sci. Total Environ.* 674, 58–70 (2019). <https://doi.org/10.1016/j.scitotenv.2019.04.101>

Reviewer Reports on the Second Revision:

Referee #1 (Remarks to the Author):

Review of Blanchfield et al. “Experimental evidence for the recovery of mercury contaminated fish”

General comments:

I appreciate the authors’ thoughtful edits and responses to my previous comments. I think the new section on elimination times in boreal fish a great addition to the paper and I like how they have incorporated more numerical (%) values characterizing the responses of sediment, water and fish. I have a few nit-picky things below that the authors can consider and generally I think the paper is now acceptable for publication. Congratulations to the authors on this nice work!

Specific comments:

Lines 47-49: There is a later paper by Streets et al. (2019, ERL) that includes releases to land and water (in addition to air) that is much higher in terms of total anthropogenic releases. If the current text is preferred then the authors should clarify these are anthropogenic releases to the atmosphere only.

Lines 67-68: I still disagree that there is little direct evidence that fish will decline following loading reductions. I think there is a lot of evidence and this study adds convincingly to that. I personally think that the review paper by Wang et al. (2019) cited here incorrectly represented the state of understanding and confused how studies attribute changes due to loading from the impacts of ecosystem change. It was not a good paper and probably a better one could be cited. I appreciate that the authors are trying to correct the misperceptions introduced by that paper. The text on lines 78-79 for example states: “it is exceedingly difficult to unambiguously assess the recovery of contaminated fish populations due specifically to Hg control measures” citing the Eagle-Smith paper - which is much better and could be substituted here.

Lines 68-70: The second half of this sentence is actually misleading now. I realize the authors are

trying to accommodate the other reviewer here but the basic point is that fish Hg concentrations respond to loading. I don't think this has to be discussed in the same sentence as the fish response paragraph. There are a lot of ecosystem models that consider these lags - e.g., the modeling conducted as part of this study and published in 2007 in PNAS. This is why earlier I recommend the authors refer to loading rather than atmospheric emissions - because these are different types of relationships but they can be characterized/quantified. The important processes for fish bioaccumulation that can affect fish responses in the context of this work are those that impact methylation/demethylation and trophic structure/bioenergetics and are discussed in later sentences. I don't think loading/watershed processes are particularly relevant here so recommend deleting the second half of this sentence. I like the rest of the paragraph now.

Referee #3 (Remarks to the Author):

I have examined the second revision and rebuttal by Blanchfield et al. I feel the authors have watered down their key statements to the point they become ambiguous, and limited in impact (impact statement removed from abstract ; time scale of impact removed in final paragraph). I understand that much is at stake to support the rapid effectiveness of the Minamata convention, but I find that the data do not warrant a timely recovery of the ELA lakes.

Abstract :

L58 "Cessation of loading emulated a 100% Hg reduction scenario that resulted in 38-91% declines in the labeled MeHg concentration of fish populations within eight years."
This main finding in the abstract is misleading, because it suggests a 38-91% recovery of fish MeHg levels, following a 100% Hg loading reduction scenario; the 38-91% declines concern only the 20% Hg wet deposition loading pathway that is discussed. The additional 80% Hg loading comes from the watershed. A 100% pollution control reduction scenario thus resulted in 8-18% overall fish MeHg declines (i.e. 38-91% of 20%).

Closing impact statement :

L201 " The most important outcome of this whole ecosystem experiment is the demonstration that a decrease in a single factor (Hg loading) has a clear and timely effect on MeHg concentrations in fish populations."

I agree with this ; but it should read '...single factor (atmospheric Hg loading)...' because the statement does not apply to watershed inputs (wrt 'timely')

L206-208 " Our findings provide strong evidence that legislation to reduce global Hg emissions would lead to the recovery of contaminated freshwater fisheries, which would in turn reduce human exposure."

The weighing of words reaches perfection here : the final impact phrase contrasts with that of revision 1 ("Hg emission controls alone will have immediate and future benefits to fish consumers") in that the time scale has been removed. It is like saying that 'grass is green'. I think the data support the following statement : "...would lead to a rapid but limited (<20%) recovery of contaminated freshwater fisheries..."

Author Rebuttals to Second Revision:

Referee #1 (Remarks to the Author):

Review of Blanchfield et al. "Experimental evidence for the recovery of mercury contaminated fish"

General comments:

I appreciate the authors' thoughtful edits and responses to my previous comments. I think the new section on elimination times in boreal fish a great addition to the paper and I like how they have incorporated more numerical (%) values characterizing the responses of sediment, water and fish. I have a few nit-picky things below that the authors can consider and generally I think the paper is now acceptable for publication. Congratulations to the authors on this nice work!

We thank the Referee for their positive comments.

Specific comments:

Lines 47-49: There is a later paper by Streets et al. (2019, ERL) that includes releases to land and water (in addition to air) that is much higher in terms of total anthropogenic releases. If the current text is preferred then the authors should clarify these are anthropogenic releases to the atmosphere only.

Perhaps there is some confusion, as it is an earlier paper by Streets and coauthors (2011 EST) that deals solely with releases of Hg to the atmosphere. The paper that we cite (reference 1; Streets et al. 2017 EST) accounts for all releases of Hg to the biosphere (land and water, as well as to the atmosphere; see Table 1) from human activities (up to 2010). We thank the Referee for pointing out this more recent paper (Streets et al. 2019 ERL), which further refines spatial and temporal (especially prior to 1850) Hg releases from human activities. The total amount of Hg released to the biosphere, ~1.5 million tonnes, is roughly the same for both studies. We have cited the newer paper instead.

Lines 67-68: I still disagree that there is little direct evidence that fish will decline following loading reductions. I think there is a lot of evidence and this study adds convincingly to that. I personally think that the review paper by Wang et al. (2019) cited here incorrectly represented the state of understanding and confused how studies attribute changes due to loading from the impacts of ecosystem change. It was not a good paper and probably a better one could be cited. I appreciate that the authors are trying to correct the misperceptions introduced by that paper. The text on lines 78-79 for example states: "it is exceedingly difficult to unambiguously assess the recovery of contaminated fish populations due specifically to Hg control measures" citing the Eagle-Smith paper - which is much better and could be substituted here.

The reviewer does not provide a reference for direct evidence that fish MeHg will decline following loading reductions, and we don't know of one. We have left this statement unchanged.

As to referencing the separate point that there are many factors that affect the transformation of Hg to MeHg and subsequent bioaccumulation, we agree with the Referee that the Wang et al. (2019) review paper is not necessarily the best one to cite, and have cited the recent Eagle-Smith paper instead, as suggested.

Lines 68-70: The second half of this sentence is actually misleading now. I realize the authors are trying to accommodate the other reviewer here but the basic point is that fish Hg

concentrations respond to loading. I don't think this has to be discussed in the same sentence as the fish response paragraph. There are a lot of ecosystem models that consider these lags - e.g., the modeling conducted as part of this study and published in 2007 in PNAS. This is why earlier I recommend the authors refer to loading rather than atmospheric emissions - because these are different types of relationships but they can be characterized/quantified. The important processes for fish bioaccumulation that can affect fish responses in the context of this work are those that impact methylation/demethylation and trophic structure/bioenergetics and are discussed in later sentences. I don't think loading/watershed processes are particularly relevant here so recommend deleting the second half of this sentence. I like the rest of the paragraph now.

We agree with the Referee that the second half of this sentence is misplaced and this part of the paragraph should focus solely on the key ecological factors that alter the concentrations of MeHg in fish in different situations and ecosystems. We have revised this sentence and included an appropriate citation with the revised text (lines 67-69). The concept in the second half of the sentence, however, is important and we have moved it rather than deleting it, as explained below.

One of the key challenges to predicting the efficacy of Hg pollution controls is being able to evaluate the relative contribution of newly deposited Hg to contemporary MeHg production. We were able to make this distinction in the METAALICUS project through our novel approach – the whole-ecosystem application of enriched isotopes of Hg – and we feel it is important to also include this statement, which is now placed after the later sentences describing how human activities can alter aquatic ecosystems and fish populations (and therefore MeHg), where it more logically fits (lines 75-76).

Referee #3 (Remarks to the Author):

I have examined the second revision and rebuttal by Blanchfield et al. I feel the authors have watered down their key statements to the point they become ambiguous, and limited in impact (impact statement removed from abstract ; time scale of impact removed in final paragraph). I understand that much is at stake to support the rapid effectiveness of the Minamata convention, but I find that the data do not warrant a timely recovery of the ELA lakes.

We appreciate the Referee pointing out that inclusion of stronger key statements would improve the manuscript, and here we now include a key impact statement in the Abstract (lines 60-62) and have revised the statement in the closing paragraph (lines 206-208).

With respect, we have a different perspective than the Referee regarding what constitutes “timely” when considering the recovery of fish MeHg concentrations from reductions in Hg loading. From our perspective, it would be hard to imagine a more rapid response by fish to loading – for some of the juvenile fish, the lake spike MeHg concentrations declined as quickly as for zooplankton! It is important to consider that there were a range of possible outcomes for this whole-ecosystem experiment. For example, what if our study had shown that eight years after ceasing additions of spike Hg, concentrations of spike MeHg in the

various fish populations did not decline, perhaps due to continued participation of past spike additions in the mercury cycle of the lake? This scenario would certainly lead us to an entirely different conclusion. Instead, we observed a very rapid decline in fish MeHg concentrations to reductions in Hg loading – including spike Hg applied to the lake (lake spike) and to the watershed (upland spike).

Abstract :

L58 "Cessation of loading emulated a 100% Hg reduction scenario that resulted in 38–91% declines in the labeled MeHg concentration of fish populations within eight years."

This main finding in the abstract is misleading, because it suggests a 38-91% recovery of fish MeHg levels, following a 100% Hg loading reduction scenario; the 38-91% declines concern only the 20% Hg wet deposition loading pathway that is discussed. The additional 80% Hg loading comes from the watershed. A 100% pollution control reduction scenario thus resulted in 8-18% overall fish MeHg declines (i.e. 38-91% of 20%).

We feel that our description of the observed decline in spike MeHg concentrations of the different fish species following a 100% loading reduction to the lake is an accurate representation of the how quickly fish *spike MeHg concentrations declined*, and is not misleading. The difference between the calculations in this comment, and the manuscript, is that the Referee is looking at the results from a 100% cessation of atmospheric loading, and we are writing from the view of a 100% cessation of loading to the lake. We think this confusion has derived from using the term “loading” sometimes when we actually meant “deposition”. The isotope *additions* were mimicking *deposition* to the watershed and to the lake surface. The term “loading” in our manuscript, however, is specifically for Hg inputs to the lake. We have gone through the manuscript and been more specific in our use of the terms “loading”, “deposition” and “addition”. To limit any confusion, we have termed the period when we added isotopic Hg to the ecosystem as “addition” (formerly “loading”) on the associated figures and tables and throughout the text.

Closing impact statement :

L201 " The most important outcome of this whole ecosystem experiment is the demonstration that a decrease in a single factor (Hg loading) has a clear and timely effect on MeHg concentrations in fish populations."

I agree with this ; but it should read ‘...single factor (atmospheric Hg loading)...’ because the statement does not apply to watershed inputs (wrt ‘timely’)

We’ve re-worded this sentence to read “...single factor (Hg loading to the lake) has a clear and timely effect on average MeHg concentrations in fish populations...” because this is what we mean. The Referee is apparently equating “atmospheric Hg loading” to direct deposition onto the lake surface. This was the major change in loading to the lake and so these two phrases are almost equivalent, but there was also a small change in watershed loading to the lake that was timely.

The Referee is confusing the response of the watershed runoff to change in deposition (not “timely”) with the response of the fish in the lake to a change in loading to the lake itself (rapid). Another reviewer had trouble with the same concept so we have carefully reworded several parts of the manuscript to correct this confusion (see our response to the preceding comment).

L206-208 " Our findings provide strong evidence that legislation to reduce global Hg emissions would lead to the recovery of contaminated freshwater fisheries, which would in turn reduce human exposure."

The weighing of words reaches perfection here : the final impact phrase contrasts with that of revision 1 ("Hg emission controls alone will have immediate and future benefits to fish consumers") in that the time scale has been removed. It is like saying that ‘grass is green’. I think the data support the following statement : "...would lead to a rapid but limited (<20%) recovery of contaminated freshwater fisheries..."

We agree that a stronger final statement is warranted. Again, as noted above, there has been some confusion between “loading to the lake”, where a reduction resulted in rapid and major reductions in fish MeHg, and “deposition”, which is what the Referee means in this comment. This was already laid out in lines 197-200 of the last version of the manuscript, but we think that the more precise terminology (discussed earlier) and new text in this revised manuscript (lines 197-199 and 203-208) will now make this even clearer.

We have retained part of the statement regarding “...demonstrating that Hg emission controls will have immediate benefits to fish consumers.” in the final sentence of the Abstract (lines 60-62). The closing sentence of the paper (lines 206-208) now states “...as these two loads decrease, the fish populations in the receiving lake will soon afterwards have lower MeHg than they would have if nothing were done, thereby reducing human exposure”. This statement brings in the rapidity of the fish response (“soon”) and links to the previous sentence regarding the key point made by this Referee, that changes in loading to the lake from directly deposited Hg (fast) and from the watershed Hg (slow) will follow different time courses in response to declines in atmospheric deposition (lines 205-206).